



# On the role of thermal expansion and compression in large scale atmospheric energy and mass transports

Melville E. Nicholls[1] and Roger A. Pielke Sr.[1]

[1]Cooperative Institute for Research in Environmental Sciences, Department of Atmospheric and Oceanic Sciences,
University of Colorado, Boulder, CO 80309, USA

*Correspondence to*: Melville E. Nicholls (Melville.Nicholls@colorado.edu)

**Abstract.** There are currently two views of how atmospheric total energy transport is accomplished. One view considers total energy as a quantity that is transported in an advective-like manner by the wind. The other considers that thermal expansion and the resultant compression of the surrounding air causes a transport of total energy in a wave-like manner at the speed of sound. This latter view emerged as the result of detailed analysis of fully compressible mesoscale model simulations that demonstrated considerable transfer of internal and gravitational potential energy at the speed of sound by Lamb waves. In this study, results are presented of idealized experiments with a fully compressible model designed to examine the large-scale transfers of total energy and mass when local heat sources are prescribed. For simplicity a Cartesian grid was used, there was a horizontally homogeneous basic state, and the simulations did not include moisture.

Three main experimental designs were employed: The first has a convective storm scale heat source. This experiment illustrates the basic process of expansion in the heated region leading to compression of the adjacent air and the subsequent formation of a Lamb wave that propagates laterally at the speed of sound. The second experiment has a continent scale heat source prescribed near the surface to represent surface heating and which includes a constant Coriolis parameter. Results indicate that after several hours considerable amounts of total energy and mass are transferred offshore at the speed of sound. This simulation is compared to one with the term in the governing equations responsible for forcing thermal compression waves omitted. There is a fairly small but nevertheless significant difference of the wind field compared to the fully compressible results. A comparison of the fully compressible simulation with a simulation without the Coriolis force illustrates the role of geostrophic adjustment in significantly influencing the total energy distribution. The third experiment has a cloud cluster scale heat source prescribed at the equator and which includes a latitude dependent Coriolis parameter. Results show considerable amounts of meridional total energy and mass transfer at the speed of the sound.

This study suggests that the current understanding of large scale total energy transfer that assumes it is accomplished solely in an advective-like manner by the winds is not correct. If so the current theory would be incomplete, and since some of the transfer might be occurring at the speed of sound this could bring into question the methodology commonly used to attribute the total energy transports to transient eddy, stationary eddy, and mean meridional circulations.

Based on these results the conceptual differences between the terms "total energy transport" and "heat transport" in a fluid are discussed. Fast moving compression waves are unlikely to be accurately simulated by most climate models since they utilize semi-implicit time differencing techniques, and this could potentially be a source of error that has yet to be evaluated.

## 1 Introduction

Satellite studies of the zonally averaged top of atmosphere (TOA) radiative fluxes show that there is a net input of energy at low latitudes and output at high latitudes (e.g., Vonder Haar and Suomi 1971). For a heat balance, there must be a poleward transfer of total energy, and fluid motions of both the atmosphere and oceans are considered to play major roles. It has been generally understood that the atmosphere and ocean can carry heat from one area to another in an advective-like manner by the winds and ocean currents. For the atmosphere this underlying assumption has been the basis for the analysis technique whereby the total energy flux across a latitude circle is decomposed into contributions by transient eddies, stationary eddies



and mean meridional circulations (Priestley 1949; White 1951a, b; Starr and White 1954; Oort and Peixóto 1983, among others). This technique has been commonly used in both observational and numerical modelling studies. However, the assumption that this is the sole mechanism responsible for total energy transfer in the atmosphere was brought into question by numerical modelling studies carried out with a fully compressible numerical model (Nicholls and Pielke 1994a and b;

hereafter NP94a and NP94b, respectively). It was found that large quantities of internal and gravitational potential energies could be transferred in a wave-like manner at the speed of sound. The basic idea is relatively straightforward: Consider for instance that as an air parcel is heated it causes the temperature and pressure of the parcel to increase (a hot air balloon would be another way to think about this). The resultant pressure gradient between the heated air and its immediate environment then drives a divergent outflow causing the heated air parcel to expand. What is usually considered the salient

point is that the expansion leads to a density decrease, and therefore makes the air parcel (or hot air balloon) more buoyant. However, at the same time, the expansion of the air parcel must also cause the air in the immediate surroundings to become compressed, and that means the pressure must increase in this surrounding air. The pressure gradient therefore works outwards, continually accelerating air in an expanding radius. The result is the generation of a compression wave, that we have termed a "thermal compression wave", which has some clear similarities to a mechanically forced sound wave.

Thermal compression waves produced in this manner are not high frequency sinusoidal waves so in this respect they differ from typical sound waves. There is a wave front that propagates at the speed of sound and behind this a region of compressed air. It is at this point that it can be seen that a transfer of energy and mass might be occurring since compressed air has more molecules per unit volume and in the mean the molecules are moving slightly faster. The perturbations of pressure, temperature and density in the large volume of compressed air surrounding the heated air parcel are extremely

small compared to those occurring in the immediate vicinity of the air parcel where relatively strong buoyancy circulations develop. Nevertheless this rapidly widening envelope of compressed air with small internal energy and density anomalies adds up over a very large volume to constitute a major reservoir of total energy and mass, and this reservoir does not stay in place once the heating ends, but propagates away rapidly at the speed of sound.

While this theory of total energy transport has been known for some time it has not yet had any significant impact on the

global scale view of how total energy transport is occurring. In this manuscript the main objective is to show that this mechanism of total energy transfer is likely to be important for large scales, and moreover to point out that this mechanism is probably not simulated very well in current global scale models. Since the perturbations associated with thermally generated compression waves are typically extremely small it might seem at first sight that their effects can be safely neglected. The case is made that inaccuracies will be produced when they are not simulated correctly on the large scale, which might not be

negligible. The manuscript is organized in the following manner: Section 2 gives an overview of the two theories. Section 3 discusses the numerical model and the design of the three experiment configurations, which consist of convective scale, continent scale, and cloud cluster scale simulations. Section 4 presents results for the three main experiments and several auxiliary experiments. In section 5 the significance of these results for understanding total energy and mass transfers are discussed as well as potential implications for climate modelling. Finally in section 6 conclusions are summarized.

**2 Overview of total energy transfer mechanisms**

**2.1 The current paradigm of total energy transport**

The derivation of the total energy equation can be found in many standard texts (e.g. Gill 1982; Peixoto and Oort 1992). It can be written in the following form:

$$\frac{\partial}{\partial t}\left(c_v\rho T + \rho g z + \rho L q + \rho\frac{u^2}{2}\right) + \nabla \cdot \left\{\left(c_v\rho T + \rho g z + \rho L q + \rho\frac{u^2}{2}\right)\boldsymbol{u} + p\boldsymbol{u} + \boldsymbol{F} - \kappa\nabla T - \mu\nabla\left(\frac{u^2}{2}\right)\right\} = 0 \qquad (1)$$

where $c_v$ is the specific heat at constant volume, $\rho$ the density, $T$ the temperature, $g$ the acceleration due to gravity, $L$ the latent heat of condensation, $q$ the specific humidity $\boldsymbol{u}$ the velocity vector, $p$ the pressure, $\boldsymbol{F}$ the radiative flux density, $\kappa$ the





thermal conductivity, and $\mu$ the viscosity. The total energy is comprised of internal energy, gravitational potential energy, latent heat energy, and kinetic energy. Note that this Eulerian conservation equation concerns the energy per unit volume.

In large-scale energy budget studies the hydrostatic approximation is made and the flux across a vertical latitudinal wall is rewritten in the (x, y, p) coordinate system (Priestley 1949). Oort and Peixoto (1983) give the following expression for the flux across a circle of latitude:

$$\int_{z=0}^{\infty} \oint \rho \left(c_p T + gz + Lq\right) v \, dx \, dz = 2\pi R_e \cos\phi \int_0^{p_0} \left(c_p T + gz + Lq\right) v \frac{dp}{g} \tag{2}$$

where the enthalpy has been introduced ($c_p T = c_v T + p/\rho$), $c_p$ is the specific heat at constant pressure, $R_e$ the radius of the earth, $\phi$ the latitude, $v$ the meridional velocity and $p_0$ the surface pressure. To derive this equation the terms involving kinetic energy, thermal diffusion and viscosity have been neglected. After deriving the above equation, Oort and Peixoto make the following statement: "Further, the integrand can be decomposed into the contributions by transient eddies, stationary eddies and mean meridional circulations in order to get a better understanding of the physical mechanisms involved in the fluxes:"

$$\left[\overline{\left(c_p T + gz + Lq\right)v}\right] = c_p[\overline{v'T'}] + c_p[\bar{v}^*\bar{T}^*] + c_p[\bar{v}][\bar{T}]$$
$$+g[\overline{v'z'}] + g[\bar{v}^*\bar{z}^*] + g[\bar{v}][\bar{z}]$$
$$+L[\overline{v'q'}] + L[\bar{v}^*\bar{q}^*] + L[\bar{v}][\bar{q}] \tag{3}$$

where over-bars represent a time average and square brackets a zonal average. The prime symbol represents the departure of the variable from the time average, and the asterisk the departure from the zonal average. Note that this decomposition tacitly makes the assumption that total energy is a quantity that is advected solely with the wind. For instance, the term $c_p[\overline{v'T'}]$ is considered a transient eddy flux of enthalpy, implying that a large-scale eddy that moved warm air poleward and cold air equatorward would be physically accomplishing a meridional total energy transport. However, studies that have conducted detailed analyses of total energy transfer using fully compressible models demonstrate that total energy can be transferred in a wave-like manner at the speed of sound, which challenges this assumption (NP94a and b; Nicholls and Pielke 2000).

### 2.2 A linearized one-dimensional solution for a thermal compression wave

When an air parcel is heated the temperature and pressure increases and the resulting pressure gradient causes it to expand. During its expansion, the air surrounding the parcel is compressed. This compression does not simply remain in situ; a wave front propagates away from the heated air parcel at the speed of sound leading to weak positive pressure, temperature, and density anomalies over an increasingly large volume surrounding the heated air parcel. The internal energy per unit volume given by $c_v\rho T$ can be rewritten using the ideal gas law $p = \rho RT$ as $c_v p/R$. Therefore a positive pressure anomaly in the region of compressed air means that the internal energy is increased in this region. In addition, the increased density means that the gravitational potential energy $\rho gz$ is increased. Therefore this mechanism can quickly lead to an increase of total energy over a very large volume surrounding the air parcel and constitutes a total energy transfer at the speed of sound. A very simple mathematical solution illustrating the basic process can be obtained for the one-dimensional linearized momentum, thermodynamic and continuity equations which allow thermal compression waves:

$$\frac{\partial u}{\partial t} + \frac{1}{\rho_0}\frac{\partial p'}{\partial x} = 0 \tag{4}$$

$$\frac{\partial p'}{\partial t} - c^2\frac{\partial \rho'}{\partial t} = \frac{\gamma p_0}{c_p T_0}Q_m \tag{5}$$

$$\frac{1}{\rho_0}\frac{\partial \rho'}{\partial t} + \frac{\partial u}{\partial x} = 0 \tag{6}$$





where $u$ is the x-component of velocity, $p'$ perturbation pressure from the base state, $\rho'$ perturbation density, $c = \sqrt{\gamma R T_0}$ the speed of sound, $\gamma = c_p/c_v$, $R$ the gas constant for dry air, and $Q_m$ the hearting rate per unit mass. The basic state values of pressure, density, and temperature are $p_0$, $\rho_0$, and $T_0$, respectively. These reduce to a single equation for perturbation pressure:

$$\frac{\partial^2 p'}{\partial t^2} - c^2 \frac{\partial^2 p'}{\partial x^2} = \frac{\gamma p_0}{c_p T_0} \frac{\partial Q_m}{\partial t} \qquad (7)$$

Consider a heating function $Q_m = U(t - t_0) Q_0 a^2/(x^2 + a^2)$ where the Heaviside unit function $U(t - t_0) = 1$ for $t > t_0$ and 0 for $t < t_0$. The equation can be solved for $p'$ by taking Laplace and Fourier transforms, solving algebraically, and then taking inverse transforms to give:

$$p'(x,t) = \frac{\gamma p_0 Q_0 a}{c_p T_0 c} \frac{1}{2} \left\{ arctan\left(\frac{ct + x}{a}\right) + arctan\left(\frac{ct - x}{a}\right) \right\} \qquad (8)$$

It is straightforward to obtain solutions to the other variables (NP94a).

Figure 1 shows the solution for a heating rate of 1 J kg⁻¹ s⁻¹, a=20 km, T₀=300 K, p₀=10⁵ Pa at t=600 s. A narrow
temperature increase occurs where the heating is large at x=500 km, but there is a very broad region of increased pressure. The solution has a wave-like character with the wave front, at around 200 km from the centre of the maximum heating, propagating at the speed of sound. There is divergence in the region of heating and convergence at the wave front. The passage of the wave front is characterized by a notable increase in pressure and relatively minor increases in density and temperature as the air is compressed.

The solution at 1200s for a heating rate that is turned off at 600s is shown in Figure 2. Turning off the heating leads to a centre that is warmed and is less dense at 1200 s with only very small pressure perturbations remaining. Two wave-like anomalies are moving in opposite directions at the speed of sound. Since the internal energy per unit volume only depends on the pressure it can be seen that the internal energy perturbations given by $c_v p'/R$, are propagating away from the heated region at the speed of sound. Even though the central region has warmed considerably there is no significant change in the
internal energy per unit volume ($c_v \rho T$) since there has been a large density decrease in the narrow heated region. Mass can be seen to be conserved since the density has increased slightly in the two wide compressed regions which are propagating away at the speed of sound. Therefore, this very simple 1D solution shows a significant redistribution of internal energy and mass occurring at the speed of sound. This is a distinctly different process than envisioned in the decomposition of Eq. 3, which has been the standard procedure for the physical attribution of energy transports used in numerous studies.

**2.3 A thought experiment demonstrating internal energy transfer is different than convective heat transfer**

Another perspective on internal energy transfer is gained by considering the thought experiment shown in Figure 3. An ideal gas in an insulated container is separated into two parts, gas 1 and gas 2, by a movable frictionless partition. Consider uniformly heating gas 1 on the left side of the container. This will cause the temperature and pressure to increase. Gas 1 will expand and the movable partition will move to the right compressing gas 2. If the heating is discontinued the partition will
come to an equilibrium position, as shown in Fig. 3b, such that the pressure in gas 2 equals the pressure in gas 1. The temperature of gas 2 will have increased due to compression, but will be considerably less than that of gas 1. Since the internal energy per unit volume ($c_v p/R$) is only a function of pressure, it will be the same in gas 2 as in gas 1. Obviously, since gas 1 now occupies a larger volume than gas 2 it will have a larger net internal energy.

The compression of gas 2 is a fast response occurring at the speed of sound. As the partition moves from left to right it
imparts momentum to the adjacent molecules in gas 2 causing a wave of compression to travel through gas 2 at the speed of sound, which reflects backwards and forwards off the right lateral boundary and the middle partition. Now consider removing the partition. If the two gases were then mixed by mild stirring this wouldn't impart significant internal energy



(stirring a gas will not increase its temperature by much), and the pressure would remain essentially the same. As far as the internal energy in a control volume in gas 2 is concerned, the internal energy increased during the compression stage. After the mixing stage, even though the control volume contains warmer air than it did at the end of the compression stage, the amount of internal energy within it has not changed since the pressure is the same. Note that the density of the gas in the

control volume that was larger than the density in gas 1 will decrease during this mixing phase as the density throughout the container becomes uniform.

Convective heat transfer is considered to be the transfer of heat from one place to another by the movement of fluids. It is common in meteorology to describe heat as being advected from one place to another, whereby it is meant that when relatively warm air moves into a region it constitutes a "heat transfer". This interpretation is at the heart of the decomposition

made in Eq. 3. From this perspective the heat transport from left-to-right in the thought experiment depicted in Fig. 3 took place in the second stage when air was mixed and the air in the control volume became notably warmer. However, as has been explained, this did not constitute a total energy transfer, which occurred earlier in stage 1 during the compression of gas 2.

The idea that convective heat transfer is not the same thing as internal energy transfer is a very simple idea as shown by

this thought experiment and shouldn't raise any red flags. Now take this one step further and propose a similar thing is actually happening in the atmosphere and that internal energy is being transferred at the speed of sound. This apparently is one step too far and this hypothesis becomes subject to incredulity.

## 2.4 Different meanings of heat transfer

It is pertinent at this stage to review three uses of the term heat transfer:

1) In thermodynamics, heat is defined as energy in transit across the boundary of a system, which does not involve work or the transfer of matter. Since heat is the flow of energy across the boundary of a system it is not correct to say it is stored in the system. However, the transfer of heat does lead to an increase of the internal energy of the system equal to the heat transferred.

2) Convective heat transfer within the body of a fluid is often defined as the transfer of heat from one place to another by the

movement of fluid elements. A moving fluid element is said to carry energy with it. This energy is often described as thermal energy, or internal energy. However, there is a problem with this interpretation as made clear by the thought experiment in Fig. 3. Transport of relatively warm air into the control volume during the mixing stage did not result in a change of the internal energy within the control volume. It led to a change in temperature, such that the mean kinetic energy of the molecules in the control volume increased, but their number decreased (corresponding to the density decreasing) so

that overall there was no net change in internal energy.

3) The expression for the change of total energy (Eq. 1) involves fluxes of enthalpy, gravitational potential energy, and latent heat energy. These summed have also been called a "heat flux".

Clearly the term "heat" is being used for different physical quantities, and this is one reason there is confusion on this issue.

Consider the gas in the container shown in Fig. 3, and suppose that the warm and cold air in a thin region at their interface

started to be mixed after removal of the partition. The turbulent eddies produced by the stirring shouldn't result in a horizontal transfer of internal energy as we have maintained. Now consider putting a brief heat source at the far left side of the container, which produced a transient compression wave that proceeded to propagate through the interface between the warm and cold air that is being stirred. As the compression wave passes from left to right through the region of eddies there would for a brief instant of time be an enthalpy flux and internal energy would consequently increase on the far right side of

the container. It would not be accurate to say that the turbulent eddies are physically responsible for that enthalpy flux just because they occupy the same space as the compression wave. In this situation the fluxes of enthalpy would not be properly represented by a decomposition into terms like Eq. 3, which would physically be attributing the transport to turbulent eddies.


It is possible that an analogous situation could occur in the atmosphere and this point will be returned to in section 5, after results of the numerical simulations are presented.

Obviously the movement of warmer air to the right side of the container and colder air to the left side is of major physical importance as are the meridional transports of warm and cold air by eddies in the atmosphere. This meridional transport in the atmosphere is playing a significant role in the overall energy balance. For instance, TOA radiative fluxes would be different without these warm and cold air advections. However the simple thought experiment discussed above suggests that the quantity total energy in Eq. 1 is not necessarily being transferred when there is advection of warm or cold air masses. The transfer of the quantity total energy is complicated to understand because the atmosphere to a reasonable approximation behaves like an ideal gas so that heat input results in expansion and work being done on the surrounding air. This work done compressing the surrounding air is fundamental to understanding why energy is quickly redistributed at the speed of sound.

## 2.5 Vertical transport of total energy

Consider a horizontally homogeneous atmosphere and low-level heat source. The heated air will expand in the vertical and work will be done not only against the atmospheric pressure as in the example discussed in section 2.2, but also against gravity. The acoustic cut-off frequency for a non-isothermal atmosphere is approximately 5 min in the troposphere and stratosphere (Beer 1974; Walterscheid et al. 2003). Shortly after the low-level heat source is turned on in a simulation an upward propagating pulse of vertical velocity develops. This pulse propagates upwards at the speed of sound and increases in amplitude as it reaches higher levels (e.g. NP94b). When it reaches 100 km above the surface after traveling for approximately 5 min, a trailing lobe with negative vertical velocity becomes evident. If there is a Rayleigh friction layer at upper levels then the wave will not reflect off the top of the model. As the simulation continues further wave-like activity of the vertical velocity occurs, but its amplitude is relatively small in the troposphere and lower stratosphere, where persistent weak upward motion prevails. Walterscheid et al. (2003) discuss results of a model simulation of acoustic waves in the mesosphere and lower thermosphere generated by deep tropical convection. They found quite a complicated behaviour with a trapped oscillation below about 80 km altitude with a period of ~5 min and a nearly vertically propagating wave with about a 3 min period above this height.

NP94b showed a low-level horizontally homogeneous heat source results in compression of the air aloft throughout a deep layer. Associated with the mean upward motion in the troposphere and lower stratosphere are a mass flux and an energy flux. From Eq. 1 the rate of change of total energy to a good approximation for a dry atmosphere is given by the convergence of the energy flux:

$$\frac{\partial}{\partial t}\left(\frac{c_v}{R}p + \rho g z + \rho\frac{w^2}{2}\right) = -\frac{\partial}{\partial z}\left\{\left(\frac{c_p}{R}p + \rho g z + +\rho\frac{w^2}{2}\right)w\right\} \tag{9}$$

where $w$ is the vertical velocity. The simulation by NP94b showed the pressure perturbation rapidly increasing with height to the top of the thin heated layer adjacent to the surface and then falling off exponentially above (Fig. 7 of that paper). Also the density decreased significantly in the heated layer and increased very slightly above but throughout a much deeper layer. These positive pressure and density perturbations aloft represent significant increases of internal and gravitational potential energy throughout a deep region above the low level heat source, and a net convergence of the energy flux given by Eq. (9) must have occurred at each level.

Later on in the simulation a turbulent boundary layer developed. The turbulent sensible heat flux was calculated and shown at low levels to a very good approximation equal to the upward total energy flux. This is an expected result since after all this is the way the enthalpy flux from the surface is in practice calculated. However, prior to the turbulent boundary layer thermals developing there was still an upward enthalpy flux. This flux didn't depend on the presence of thermals, only on the vertical expansion of the low level air.



As part of this current modelling study we have conducted a simulation with a model top at 140 km with a 20 km deep Rayleigh friction layer. The results remain similar to NP94b showing a significant increase of the total energy up to the lower stratosphere in response to a low-level heat source. Therefore if one wanted to calculate the net heat input, and also calculate the increase of the perturbation energy field in response to this net heat input, then the calculation would have to

integrate vertically through a large depth, into the stratosphere, to get a reasonable agreement between them.

**2.6 Lamb waves**

Solutions to the linearized equations for a stratified fluid show the existence of three types of waves known as acoustic waves, Lamb waves, and gravity waves (e.g. Gill 1982). The restoring force for acoustic and Lamb waves is compression, and for gravity waves is buoyancy. The properties of Lamb waves have been discussed by numerous researchers (e.g. Lamb

1910, 1932; Taylor 1936; Lindzen and Blake 1972). A feature of this wave is that for an isothermal atmosphere their velocities are parallel to the surface, so the vertical velocity is zero.

The vertical structure of Lamb waves for an isothermal atmosphere is given by:

$$u' \propto exp\left(\frac{(\gamma-1)z}{\gamma H_s}\right) \tag{10}$$

$$p' \propto exp\left(\frac{-z}{\gamma H_s}\right) \tag{11}$$

$$\rho' \propto exp\left(\frac{-z}{\gamma H_s}\right) \tag{12}$$

where $H_s=RT/g$ is the scale height. Since $\gamma-1$ is greater than 1 it can be seen that the perturbation horizontal velocity increases with height, whereas the perturbation pressure and density decrease with height. NP94b modelled a two-dimensional sea-breeze circulation produced by a surface heat source that was turned off after an hour. The solution at two hours showed Lamb waves had moved off shore on either side of the land surface.

Nicholls and Pielke (2000) using a fully compressible version of the Regional Atmospheric Modelling System (RAMS) modified to include sources and sinks of water vapour simulated an isolated convective storm. They found a Lamb wave emerged from the storm, which was trailed by slower moving gravity wave modes. An analysis of internal and gravitational potential energies showed that the net total energy within the Lamb wave, propagating away at the speed of sound, was approximately equal to the net latent heat release that had taken place in the convective storm. It was noted that because the

Lamb wave propagated in the horizontal plane it had a two-dimensional character. For instance, whereas a heat source in one dimension creates a compression wave with only a positive pressure perturbation as in Fig. 2, in two dimensions a leading positive pressure anomaly is trailed by a negative pressure anomaly (Fig. 2 of Nicholls and Pielke 2000). This geometric effect on the shape of sound waves is well known (e.g. Lighthill 1978).

These results suggest that thermally generated Lamb waves are likely to be ubiquitous in the atmosphere, and moreover

that they play a significant role in horizontal transfers of total energy and mass. Unlike for acoustic waves there is no cut-off frequency, so very low-frequency Lamb waves can be produced by convective forcing. Nishida et al. (2014) studied background Lamb waves in the Earth's atmosphere. They found evidence of Lamb waves from 0.2-10 mHz based on array analysis of microbarometer data with root mean square amplitude (rms) of about 0.15 Pa. In their assessment of excitation mechanisms they emphasized turbulence as the probable source for the background Lamb waves. However, it is possible that

thermally generated Lamb waves caused by latent heat release in storms could have contributed based on the results of Nicholls and Pielke (2000).

**2.7 To what extent can these thermally generated disturbances be considered waves?**

While we use the term thermal compression waves to refer to large regions of compressed air moving at the speed of sound that are generated by heat sources it is important to recognize that they don't satisfy every criterion usually associated with





waves. For instance, waves are often considered a periodic disturbance of the particles of a substance and that after their passage there is no resultant movement of the particles. Sometimes a wave is defined as a process that transports energy without transporting mass. In contrast, the pressure pulse in Fig. 2 is not periodic and its passage leads to a net movement of the fluid particles. There is clearly a mass transfer in this process.

While it might be considered stretching the definition of a wave to refer to these thermally generated disturbances as waves, there is little doubt that a thermally generated disturbance like the one that propagated at the speed of sound in the simulation of a convective storm by Nicholls and Pielke (2000) would be regarded as a wave-like feature if it was detected by a microbarometer array.

**2.8 Thermally generated gravity waves**

In the numerical experiments discussed later in this paper the prescribed heat sources also generate gravity waves as well as compression waves, so in order to get a better understanding of these simulations a brief review is given in this section. Bretherton and Smolarkiewicz (1988) drew attention to the link between gravity waves and compensating subsidence in the environment surrounding a convective cloud. The spreading gravity wave response to a developing cloud was investigated using a two-dimensional simulation. They found that a buoyancy source caused vertical displacements in the air in a

widening region around the cloud which for this two dimensional framework made the buoyancy of the environment approximately equal to the buoyancy in the cloud.

Nicholls et al. (1991) derived simple analytical two-dimensional solutions to the linear hydrostatic Boussinesq equations for an atmosphere at rest with prescribed heat sources and sinks. For a case with an idealized rigid lid and a deep heat source, represented by a half-sine wave in the vertical, the thermally generated buoyancy circulation was characterized by upward

motion in the heated region with outflow aloft and inflow at low levels. The outflow and inflow expanded rapidly on either side of the heat source. The leading edges of these expanding circulations were deep fast-moving wave-like pulses of subsidence. Therefore, the subsidence compensating the central upward motion did not occur continually over broad regions on either side of the heat source, but had a distinct horizontally propagating character. The propagating subsidence regions caused adiabatic warming, adjusting the environmental potential temperature towards the perturbed values at the heated

centre for this two-dimensional framework. Response to a thermal forcing profile more typical of a mesoscale convective system (MCS) having a stratiform region was also examined for a rigid lid. In this case a deep rapidly propagating circulation like the one previously discussed was superimposed on a slower propagating circulation characterized by a mid-level inflow and upper- and lower-level outflows. This second slower moving mode had a cool potential temperature anomaly at low levels and a warm potential temperature anomaly aloft. The leading pulses of vertical motion had upward

motion at low levels and downward motion aloft. The speed of the modes is given by

$$c_n = \frac{NH}{n\pi} \tag{13}$$

where N is the Brunt-Väisälä frequency, H the height of the rigid lid, and n the wave number of the vertical heating with a vertical structure $\sin(n\pi z/H)$, where $z$ is height.

The two-dimensional solution for a semi-infinite region, without a troposphere/stratosphere interface, shows considerable

differences of the low-level fields in some respects (Pandya et al. 1993). In particular, the magnitude of the subsidence is substantially reduced, and occurs over a much broader region. Moreover, the axis of the peak vertical velocity in the low-level subsidence region is no longer vertically aligned, but strongly tilted. Nevertheless, adiabatic warming behind the broader wave front still gradually approaches the values at the heated centre. Another factor to consider is that in the real atmosphere there is increased stability above the tropopause, which partially reflects waves and to some extent increases the

similarity with the rigid lid solution. A two-dimensional squall line simulation by Nicholls (1987) showed a structure similar to the analytic solution for the first mode during the early stage of development. The deep convective heating extending to



the top of the troposphere produced a deep overturning circulation with surface mesolows growing laterally away from the centre of the convection at a rapid pace. For the first deep convective mode that extends throughout the depth of the tropical troposphere, H is approximately 15 km and taking N=0.01 s$^{-1}$ gives a horizontal propagation speed of 48 m s$^{-1}$. For the second mode, the speed is 24 m s$^{-1}$. These speeds are much faster than typical atmospheric motions.

Mapes (1993) postulated that higher order modes of the heating profile in a MCS may cause upward displacements at low levels in the nearby atmosphere, thus favouring the development of additional convection nearby. He also emphasized that the wave-like disturbances are not ordinary gravity waves, and pointing out their similarity to tidal bores in water, referring to them as buoyancy bores. There has been considerable amount of research since these earlier studies that has examined their role in convective systems (e.g. Pandya et al. 2000; Nicholls and Pielke 2000; Haertel and Johnson 2001; Fovell et al.

2006; Tulich and Mapes 2008; Bryan and Parker 2010; Adams-Selin and Johnson 2013). Inclusion of planetary rotation confines the compensating subsidence and adiabatic warming caused by deep convection to a finite distance, measured by the Rossby radius of deformation (e.g. Liu and Moncrieff 2004).

The studies by NP94a and b, were motivated by the finding that thermally generated gravity waves simulated with the standard version of RAMS (without the terms on the right side of Eq. 15 discussed in the next section) were not transferring

significant amounts of total energy, and moreover that there was a considerable discrepancy between the heat input and the net total energy within the domain. This led to the conclusion that thermally generated compression waves, which were eliminated from the RAMS model equations, might be the explanation for the missing energy.

## 3 Numerical model

RAMS is a nonhydrostatic numerical modeling system comprising time-dependent equations for velocity, non-dimensional

pressure perturbation, ice-liquid water potential temperature (Tripoli and Cotton 1981), total water mixing ratio and cloud microphysics (Pielke et al. 1992; Cotton et al. 2003). In this study simulations are conducted without moisture. The model uses a time-splitting procedure which provides numerical efficiency by treating the sound wave modes separately (Derickson 1974; Klemp and Wilhemson 1978). To obtain the most accuracy for these simulations very small time steps were used and the time splitting scheme was not employed. The model uses a similar approach to Klemp and Wilhelmson (1978) who

utilize the Exner function given by

$$\pi = \left(\frac{p}{p_0}\right)^{\frac{R}{c_p}} \tag{14}$$

The prognostic equation for the Exner function is written in tensor form as

$$\frac{\partial \pi}{\partial t} + \frac{c^2}{c_p \, \overline{\rho}\overline{\theta}_v^2} \frac{\partial\left(\overline{\rho}\overline{\theta}_v u_j\right)}{\partial x_j} = -u_j \frac{\partial \pi}{\partial x_j} + \frac{R\pi}{c_v} \frac{\partial u_j}{\partial x_j} + \frac{c^2}{c_p \, \overline{\theta}_v^2} \frac{d\theta_v}{dt} + D_\pi \tag{15}$$

where $u_j$ (j=1, 2, 3) are the velocities u, v, w, respectively, $\theta_v$ is the virtual potential temperature, and $D_\pi$ represents turbulent mixing. Bars over variables refer to the initial state which is a function of z only. Klemp and Wilhelmson (1978) note that the terms on the right side of this equation appear to have little influence on processes which are of physical interest in the

cloud model. They coded their model so that these terms could be included or omitted. They found that omission of these terms can allow small amounts of mass to be added or removed from the domain which in turn affects the mean level of the pressure. Furthermore they liken this behaviour to that occurring in the anelastic system where non-uniqueness of the Poisson solution requires an arbitrary specification of the mean pressure. For previous studies we have used the density instead of the Exner function (NP94a and b) and a different form of Eq. 15 which includes a moisture source or sink

(Nicholls and Pielke 2000). For this study we use an equation similar to Eq. 15 which was also used by Medvigy et al. (2005) for a study with RAMS that examined mass conservation. The advantage of using this form of the equation is that inclusion or not of the third term, with the material derivative of virtual potential temperature, acts as a switch for allowing





or not allowing thermally generated compression waves. Therefore the effects of thermally generated compression waves can be more readily evaluated.

Preliminary experiments that examined large scale transfer of total energy showed distortions of the compression waves occurring at large distances from the source. In order to improve accuracy the model code was modified for double
precision.

## 4 Description of experiments

There are three main experiments conducted for this study and several auxiliary experiments. The first shows the response to a deep convective scale heat source. The fields are portrayed in detail showing the air expanding in the heated region creating a compression wave in the surrounding air and shortly after the development of a buoyancy driven circulation. An
auxiliary experiment examines the effect of not including the first three terms on the right side of Eq. 15, which eliminates compression waves. The second main experiment has a continent scale heat source prescribed near the surface in a square region and includes a constant value of the Coriolis parameter specified for latitude of 45 degrees. The purpose is to show that there is a considerable transport of total energy and mass to large distances offshore. An auxiliary experiment examines the effect of not including compression waves. Another auxiliary experiment examines the effect of turning off the Coriolis
force. The third main experiment has a heat source located at the equator which is the size of a large cloud cluster and includes a Coriolis force that varies with latitude. The purpose of this experiment is to examine the resultant meridional transfer of total energy and mass. An auxiliary experiment examines the effect of turning the heat source off after a short period so that the compression wave can separate from the more slowly propagating buoyancy driven circulation. Table 1 lists the experiments.

The model is configured with 101 vertical levels and a vertical grid increment of 500 m. Below the model top which is at 50 km there is a 10 km Rayleigh friction layer in order to damp upward propagating waves. There are 161 horizontal grid points both in the x-direction and y-direction for all experiments. For the first experiment the horizontal grid increment is 4 km giving a domain width of 640 km. The heating function is

$$Q = Q_0 \sin\left(\frac{z}{H}\right) \frac{a^2}{(a^2 + x^2 + y^2)^2} \tag{16}$$

where $Q_0$ is the magnitude of the heat source, H=10 km, and $a$=12 km. The simulation is run for a 15 min period. For the
continent scale heat source simulation the horizontal grid increment is 200 km giving a domain width of 32000 km. The heat source is horizontally homogeneous for a square region in the centre of the domain with sides of length 4000 km. In the vertical the heat source is a quarter cosine between the surface and 1 km. In effect this means the heating source is applied at the first thermodynamic grid point above the surface, which is at 250 m, and at the second thermodynamic grid point above the surface, which is at 750 m. The simulation is run for a period of 9 h. For the third experiment the horizontal grid
increment is 100 km giving a domain width of 16000 km. the heating function has the same form as Eq. 16 except $a$=500 km. The simulation is run for a period of 5 h. For Experiment 3B the heating is turned off after 2h.

The initial temperature profile is the mean Atlantic hurricane season sounding of Jordan (1958) up to 50 hPa, and above this level the U. S. Standard Atmosphere profile was used (e.g. Wallace and Hobbs 1977). The simulations were dry.

## 5 Results

### 5.1 Convective scale heat source

Figure 4 shows x/z vertical sections through the center of the domain of the heating rate, temperature perturbation, pressure perturbation, x-component of velocity u, and vertical velocity w at 10 s. The heating causes the temperature and pressure to





increase as would be expected and the pressure gradient is already starting to drive an outflow as can be seen in the velocity fields. Note the pressure is starting to increase at the surface as the downward propagating wave front starts to reflect off it.

Figure 5 shows vertical sections of the density perturbation and the pressure perturbation at 20 s. The density, which is not a predicted variable, is diagnosed using the equation of state. The red/blue color scheme in this figure has a non-linear scale

in order to portray very small and very large values on the same figure. It can be seen that the density has started to decrease significantly in the core of the heated region as the air expands, while in the immediate surroundings of the heated region the air is compressed. The pressure field now shows a maximum at the surface. The higher pressure can be seen to be spreading out beyond the heat source.

Figure 6 shows vertical sections of the density perturbation, pressure perturbation and vertical velocity at 40 s. The

buoyancy created by the decreased density is starting to produce a significant updraft in the troposphere. Above this the updraft extends to a height of about 18 km. For a speed of sound of 320 m s$^{-1}$ the distance a wave front would travel in 40 s is around 12 km. Since the heating peak is at 5 km the compression wave front reaching a height of ~18 km is reasonable. At this time the surface pressure directly beneath the heat source has started to become negative due to lateral expansion causing the vertical column of air above the surface to weigh less.

Figure 7 shows vertical sections of u at 80 s and w at 120 s. At 80 s there now an inflow at low levels and an outflow in the upper troposphere associated with the buoyancy driven circulation. At 120 s in addition to the strong updraft in the mid-troposphere there is subsidence beginning to develop in the immediate surroundings, which is again a feature of the buoyancy driven circulation.

Figure 8 are vertical sections of w at 60 s and 90 s, which extend to the top of the domain at 50 km in order to portray the

vertical propagation of the compression wave. The wave front quickly propagates upwards at the speed of sound. It can be seen that the magnitude of vertical velocity in the upward propagating wave increases at it travels upwards. A second maximum is evident at 90 s, which is due to reflection of the downward propagating wave front seen at Fig 4e at the surface.

Figure 9 shows vertical sections of the density perturbation and horizontal sections of the surface pressure perturbation at 5 min and 10 min.  At 5 min the positive density perturbation at the leading edge of the compression wave has reached 100

km from the center of the heat source. The density perturbation decreases with height approximately as would be expected for a Lamb wave (Eq. 12). There is a ring of high pressure at the surface. By 10 min the compression wave has propagated to approximately 200 km from the storm. Interestingly the high pressure anomaly is concentrated in a narrow ring at the leading edge and so has a different shape than for the 1D thermal compression wave solution, shown in Fig. 1. This is likely to be mainly a geometric effect since thermally generated Lamb waves have a two-dimensional character when propagating

in three dimensions away from a localized heat source as discussed in section 2.6.

The features of the buoyancy driven simulation are mainly illustrated in Figure 10, which shows fields at 15 min. Even for this simple constant heat source the response is quite complicated. The warm region at the centre of the heat source is surrounded by an annulus of warm air. There is a v-shaped region of cool air aloft. A low pressure anomaly occurs at the surface and a high pressure anomaly at the top of the heat source. The density perturbation is similar in shape to the potential

temperature perturbation although of opposite sign. As expected there is a strong inflow at low levels and outflow at the top of the heat source. Upward motion is in the center and compensating subsidence in the surrounding air. There is a wave-like character to the buoyancy circulation, such that the ring of subsidence is propagating away from the heat source with a speed of approximately 45 m s$^{-1}$. This subsidence results in adiabatic warming. This propagating character and structure has similarities to the highly idealized solutions discussed in Nicholls et al. (1991) for the deep rapidly propagating convective

mode. Fig. 10 e shows that by 15 min the leading edge of the compression wave has almost propagated to 300 km and is considerably less in amplitude than at 10 min (Fig. 9d).

Figure 11 shows results for the convective scale heat source with the terms on the right side of Eq. 15 omitted (Experiment 1B) at 10 s. Comparing with Fig. 4 there are significant differences. The potential temperature perturbation is





significantly less, which is probably due to the upward motion at the center of the heat source leading to some adiabatic cooling, which does not exist in the case of the fully compressible simulation. There is no expansion of the air and the pressure immediately starts to fall at the surface. Figure 12 shows the density perturbation at 40 s, u at 80 s, and vertical velocity at 120 s, that can be compared with the corresponding frames in Fig. 6 and 7 for the compressible simulation. The

density perturbation is larger in magnitude than for the fully compressible case. Also, the inflow is stronger and the outflow weaker, which is consistent with an outflow velocity associated with expansion of the heated air for the compressible case superimposed onto the buoyancy driven circulation. This simulation at 120 s has developed a more extensive subsidence region surrounding the updraft. Figure 13 at 15 min can be compared to the compressible case in Fig. 10. Despite the differences noted early in the simulations, at this time the results are virtually identical. One discernable difference is that the

pressure perturbation at the top of the heat source for the compressible case (Fig. 10b) is slightly higher (Fig. 13b). These results corroborate the conclusion by Klemp and Wilhelmson (1978) that the terms on the right side of Eq. 15 are not important for cloud modeling, at least if one is not interested in the very low amplitude compression waves that are generated. However, it shall be seen for the continent scale heat source that differences are more pronounced. Also as Klemp and Wilhemson (1978) noted, for convective cloud simulations there can be a small amount of mass change within the

domain, which is evident from Fig. 13b since the mean surface pressure has been reduced (assuming that the pressure field does not deviate that much from hydrostatic). As will be discussed in section 5, mass changes could be more significant if diabatic heat sources are large, or if they occur throughout the whole domain which can be the situation for radiative forcing.

## 5.2 Continent scale heat source

In this section the response to horizontally homogeneous heating at the surface of a square continent is considered. Figure 14

shows results at 9 h for Experiment 2A, which has the Coriolis force included. The surface potential temperature over the heated land surface has increased by approximately 6 K. The surface pressure has decreased considerably over the land and has increased by ~0.16 hPa over the adjacent ocean (note a small colour bar interval has been chosen for Fig. 14b so that the magnitude of the surface pressure drop over land is not shown). A vertical section through the centre of the continent of the density perturbation (Fig. 14c) shows a large decrease near the surface (note that due to the small colour bar interval the

large decrease over land is not coloured) and a smaller amplitude increase above and over the adjacent ocean. The total energy perturbation consisting of internal, gravitational potential and kinetic, have increased in a large dome-shaped region except for a decrease in a thin layer next to the land surface (Fig. 14d). It can be seen that the total energy has increased significantly into the lower stratosphere, but not at higher levels, which is consistent with the discussion of section 2.5. Fig 14e shows the field of vertically integrated total mass change (this can be considered the change in a square metre oblong

column extending from the surface to the top of the domain). The mass has decreased over the land area and has increased over the surrounding ocean area. Fig. 14f shows a vertical section of the zonal velocity u. At the west and east edges of the land surface there are low-level inflows and just above are return flows. These are very crudely resolved sea breezes. There is a deep layer of weak offshore flow aloft. While some of this offshore flow is associated with the return flow of the sea-breeze, a part of this flow is due to expansion of the air occurring above the continent. Below the Rayleigh friction layer

there is a wave-like structure aloft. These are propagating at the speed of sound and the leading edge of the first anomalies have passed beyond the west and east sides of the panel at -10,000 km and 10,000 km, respectively (note the model domain size is much larger than the panel size shown in this figure). This wave-like appearance of the velocity field aloft is primarily a geometrical effect as discussed in section 2.6 and is not nearly so evident in a two dimensional simulation (not shown). It has ramifications for the fluxes of energy and mass off the continent as will be discussed shortly.

Figure 15 zooms in on the west side of the domain to show details of the sea-breeze circulation. There is a dome of positive pressure anomaly over a surface negative pressure anomaly. The total energy perturbation field in Fig. 14d mainly reflects these pressure perturbations. The y-component of velocity v is negative near the surface because the Coriolis force



turns the inflow winds to the right, whereas the opposite occurs in the overlying sea-breeze return flow. Fig. 15d shows a horizontal section of a larger portion of the domain at z=15.25 km illustrating an outflow aloft above the shoreline and a weak anti-cyclonic flow of several centimetres per second above the ocean that looks like it might be close to a gradient wind balance since it encircles the dome of high pressure.

Figure 16 shows time series of the summed energies throughout the whole domain and the heat input. The gravitational potential energy at any time is almost exactly two fifths of the internal energy as would be expected for a hydrostatically balanced circulation where they would differ by a factor of $R/c_v$. The total energy in the domain is the same as the heat input until seven hours, after which it becomes slightly larger. It is not clear why this small discrepancy occurs. The kinetic energy is shown on a different panel because it is so much smaller in magnitude than the internal and potential energies.

The summed energies shown separately for the land and ocean in Fig. 16d show an interesting behaviour. The internal energy initially increases quickly over land. As shown in Fig. 3 of NP94b the pressure initially increases in the heated layer adjacent to the surface, but there is an offshore mass flux as the air expands causing the surface pressure to begin to fall a short distance inland. This outflow associated with expansion of air works its way inland at the speed of sound. It takes a little under two hours for this outflow to reach the centre of the domain, which for this simulation is 2000 km from the

shoreline. There is an upward mass flux over land and an upward total energy flux (Fig. 13 of NP94b illustrates an upward sensible energy flux, or enthalpy flux, caused by a low-level heat source), which begins immediately the heating is turned on. However, at the centre of the domain the air does not expand laterally until the inward propagating outflow reaches the centre, so to begin with there are only upward fluxes, not lateral fluxes. The offshore lateral fluxes of total energy at the shoreline while small to begin with, increase quite rapidly, such that after two hours the total energy over land no longer

increases despite the heat input and actually starts to decline. This increase of the offshore flux appears to be because all the air across the whole continent is undergoing lateral expansion by two hours whereas initially it was only air near the coast. Fig. 14f shows a wave train in the zonal velocity field (u) at upper levels. The regions in the wave train, with a flow towards the land, started to develop over the continent at 3 h and by 5 h there was a significant onshore flow above 25 km. This coincided with the total energy over the land increasing again. By 9 h there is again a deep region of offshore flow at the

shoreline (Fig. 14f), but not as strong as at 2 h, and the total energy over land is continuing to increase.

    The time series of the mass changes over land and ocean shown in Fig. 16d are mirror images of one another. This indicates there is reasonably good mass conservation. The rate of mass change is slow to begin with, strongest between 2-5 h, and then changes at a moderate rate. This behaviour would be expected from the evolution of the normal component of velocity (u) at the shoreline discussed in the previous paragraph.

Figure 17 shows results at 9 h for Experiment 2B without thermal compression waves and can be compared with Fig. 14. The surface potential temperature change is identical. There is a pressure decrease over land, but there no increase over the surrounding ocean. The density decreases where heating occurs, but does not show a significant increase aloft or over the ocean. The vertical section of pressure perturbation does not show a pronounced dome. There is a mass decrease over the land, but not over the ocean so that mass is not being conserved. While a sea-breeze is evident there is not a deep offshore

flow at the shoreline.

    Figure 18 zooms in on the west side of the domain for Experiment 2B similarly to Fig. 15. The pressure drop over the land is considerably larger for this simulation compared to the fully compressible result. For the compressible case there is a larger density increase aloft comparing Fig. 14c with Fig. 17c, so that the weight of a column of air above the surface is greater which is why the pressure fall is not so much. For this simulation the inflow is larger whereas the return flow aloft is

less when compared to the compressible simulation. This can be explained by the expansion of air over the continent for the compressible simulation contributing an offshore component to the flow (Fig. 14f), which at low levels can be thought of as superimposed on the buoyancy driven sea-breeze circulation. Since the onshore flow is larger for this simulation the Coriolis force causes the meridional component of the wind to be larger in magnitude as well. These differences of the wind speeds



are approximately 15%, so while not large they are significant. Fig. 18d shows the flow aloft above the shoreline is only about one third of the strength as occurred for the fully compressible simulation (Fig. 15d). Moreover there is no anti-cyclonic flow at a large distance offshore.

Time series for the total energy in the domain and mass changes over the land and ocean for Experiment 2B are shown in

Figure 19. The total energy in the whole domain decreases slightly with time even though there is a large heat input showing there is not energy conservation. The mass in the domain decreases linearly with time and this is mainly over the land mass. The sea-breeze circulation is tending to bring some mass onshore, which results in a decrease over the ocean.

Figure 20 shows a horizontal section of the surface pressure perturbation and a vertical section through the centre of the domain of the total energy perturbation for Experiment 2C, which does not include the Coriolis term. Comparing Fig. 20a

and b, with Fig. 14b and d, it can be seen that the inclusion of the Coriolis force radically alters the pressure field and distribution of total energy. Far less total energy occurs over the land surface for this simulation and it is being redistributed much further offshore. Time series for this simulation shown in Figure 21 illustrate how much larger the total energy is over the ocean than over the land at the end of the 9 h simulation. There is also significantly more mass transported offshore for this experiment than for Experiment 2A (cf. Fig. 16d). For sea-breeze circulations the horizontal extent is comparable to the

Rossby radius of deformation, which can be considered approximately the buoyancy frequency N times a height scale for the convective boundary layer divided by the Coriolis parameter, with modifications due to surface friction (e.g. Rotunno 1983; Dalu et al. 1989; Drobinski and Dubos 2009). The sea-breeze is only poorly represented in these current simulations, however it can be seen that geostrophic adjustment associated with the fast propagating Lamb wave is clearly occurring over a much wider horizontal extent (Fig. 15d).

**5.3 Equator heat source**

In this section the response to a heat source with the scale of a cloud cluster and which is located at the equator is examined (Experiments 3A and 3B). Figure 22 shows a vertical section through the centre of the heat source and several panels of simulated fields at 1 h. There are quite a few similarities to the convective-scale heat source. A compression wave is generated which at this early stage is quite symmetrical as can be seen by the surface pressure perturbation. Figure 23, with

vertical sections that only go to a height of 20 km, shows that by 5 h a significant asymmetry has developed as the compression wave has travelled far enough away from the equator to become influenced by the Coriolis force. The potential temperature has increased to just over one degree in the centre of the heat source. The warm region has widened due to the compensating subsidence (Fig. 23f). The compression wave front has reached about 6000 km from the source as can be seen in Fig. 23b, c and d. It appears that as the wave front reaches higher latitudes the Coriolis force acts to turn the weak outflow

to the right in the northern hemisphere and to the left in the southern hemisphere resulting in a convergence of mass to the east of the heat source and a divergence of mass to the west, and this results in an asymmetrical surface pressure field. For instance, Fig. 23d shows that the meridional component of the outflow is stronger to the west of the heat source than to the east. The height-meridional vertical section of the density field shown in Fig. 23c, which has a very small colour bar interval to portray the compression wave, indicates mass redistribution towards the polar-regions. Figs. 23e and f indicate that the

leading edge of the thermally generated buoyancy circulation is propagating at a speed of ~45 m s$^{-1}$ in the meridional direction.

Figure 24 focuses on the energy and mass changes at 5 h for Experiment 3A. The height-meridional section of the internal energy perturbation shows that the leading pulse of the compression wave is a region of enhanced internal energy. This field is identical to the perturbation pressure except for the multiplication factor of $c_v/R$. There are much larger positive and

negative perturbations in the region of the buoyancy driven circulation, but these tend to cancel one another. The density perturbation at the leading edge of the compression wave, shown in Fig. 23c, results in a positive gravitational potential energy perturbations that are maximized in the upper troposphere and lower stratosphere. The fields of vertically summed



internal and gravitational potential energy perttubations show maxima to the east of the source, but clearly there has been a significant meridional transfer towards the polar-regions. The changes at the centre of the heat source are relatively small. Therefore the heat source has not caused a significant in situ increase of total energy when the vertical sum is considered. Figure 24e shows a height-meridional section of the total energy perturbation, which goes to the top of the domain. The

largest anomaly in the compression wave occurs just over 5000 km from the heat source. The major anomalies occur in the region of the buoyancy driven circulation, but vertically integrated they tend to cancel. Fig. 24f shows the mass has decreased in the region of the buoyancy driven circulation where there has been net lateral expansion of the air, whereas it has increased in the surrounding environment. The mass changes in the surrounding environment are much smaller in magnitude, but occur over a much larger area.

Time series of the energies and heat input for the whole domain are shown in Figure 25. Again the potential energy can be seen to be two-fifths of the internal energy to a good approximation. For this simulation the total energy is slightly less than the heat input at the end of the five-hour simulation, demonstrating reasonable energy conservation.

Meridional fluxes that have been summed in the vertical are shown in Fig. 26. The enthalpy flux is quite strong at the leading edge of the compression wave and is towards the poles. Interestingly the flux is towards the equator in the region

occupied by the buoyancy driven circulation. The strong low-level flow towards the heat source (Fig. 23e) and the higher pressure at low levels compared to aloft is causing this net equatorward flux. The potential energy flux on the other hand shows the opposite behaviour in the region of the buoyancy driven circulation. Since the potential energy is small at low levels, the strong outflow in the upper troposphere dominates the transport. The total flux illustrates that a significant meridional transport away from the equator is occurring between the heat source and the leading edge of the compression

wave. Fig. 26d demonstrates that there is also a broad region of meridional mass transport away from the equator, except for a small region very close to the heat source.

Results for Experiment 3B at 5 h, which is identical to Experiment 3A except the heat source is turned off after 2 h are shown in Figure 27. There is no longer a warm potential temperature perturbation at the centre, but instead a warm air anomaly in a ring that surrounds a cool anomaly at the centre. This warm ring is propagating as subsidence causes adiabatic

warming at its leading edge and ascent causes adiabatic cooling at its trailing edge. In between the downdraft and updraft the winds are towards the centre in the lower troposphere and away from the centre in the upper troposphere. So there is an overturning circulation that propagates through the environment away from where the heat source was located in a wave-like manner, which has similarities to the idealized linear two-dimensional rigid lid solution for a pulse forcing function discussed in Nicholls et al. (1991). This thermally generated gravity wave, or buoyancy bore (Mapes 1993) propagates

significantly slower than the thermally generated compression wave, so that by this time the two wave types are starting to become separated. The Lamb wave is evident in in the density perturbation field shown in Fig. 27d and it can been that it has a leading lobe with a positive anomaly and a trailing lobe with a negative anomaly, which we have mentioned previously is a geometric effect due to the Lamb wave having a two-dimensional character for a localized compact source in three dimensions. The idealized one-dimensional solution for a thermal compression wave shown in Fig. 3 on the other hand has a

simpler structure and does not exhibit a leading positive lobe trailed by a negative lobe.

Energies and mass changes for Experiment 3B is shown in Fig. 28. The leading lobe of the Lamb wave has positive internal and potential perturbation energies whereas as the trailing lobe has negative perturbation energies. Fig. 28d shows the negative values of the vertically summed total energy perturbation in the trailing lobe are as large in amplitude as the positive values in the leading lobe, but they occupy a much smaller area so there is still a net positive total energy in the

Lamb wave. Moreover by the time the negative lobe was to reach the same distance from the source as the positive lobe it would have decreased significantly in amplitude. At the centre the vertically summed total energy perturbation is slightly negative. Positive values of vertically summed total energy perturbation are only found at over 3000 km from the centre of where the heat source was located. The vertically summed mass change has a large negative value in the area occupied by

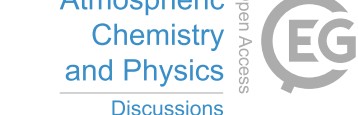

the thermally generated gravity wave, weak negative values in the trailing lobe of the compression wave and weak positive values in the leading lobe. This indicates that a significant redistribution of mass has occurred at the speed of sound.

**6 Discussion**

The current paradigm of large-scale meridional transport of internal and gravitational potential energies assumes it is accomplished solely in an advective-like manner with the winds. The results of this study suggest that significant transports could be occurring at the speed of sound. The simulations in this study are highly idealized, but it seems unlikely that this transport mechanism wouldn't be occurring in the real atmosphere. To what degree is currently unknown. The decomposition of Eq. 3 has been the standard procedure for many years that attributes the meridional fluxes to advective-like transports by transient eddy, stationary eddy, and mean meridional circulations. This study has not considered energy transfer in the presence of eddies that transport warm air poleward and cold air equatorward. However, it could be imagined for the equator heat source experiment that there was a large-scale eddy at mid-latitudes transporting warm air poleward and cold air equatorward and that a Lamb wave produced by a rapidly intensifying cloud cluster or tropical cyclone near the equator propagated polewards through it. For instance, in a similar manner as portrayed by Figs. 22f and 23b that show the rapidly propagating surface pressure perturbation. This is analogous to the scenario discussed in section 2.4. Presumably the fast moving Lamb wave wouldn't be too influenced by the much slower winds of the large-scale eddy.

Suppose for the sake of argument that the large-scale eddy wasn't producing meridional sensible and potential energy fluxes prior to the arrival of the Lamb wave. As the Lamb wave passed through the eddy there would be poleward meridional sensible and potential energy fluxes. The flux would be associated with very small increases in the meridional velocity, pressure, density and temperature. If Eq. 1 was used the enthalpy flux could be either calculated using the density and temperature, or from the pressure after substitution from the ideal gas law (ignoring the effects of moisture for simplicity). It is perhaps worth mentioning that these perturbations associated with the passage of the Lamb wave are very small and would be extremely difficult to measure from atmospheric observations, although in a modelling study this would be feasible. Alternatively the right side of Eq. 2 could be used to calculate the energy transport. In this case the bounds of the integral vary since the surface pressure rises as the Lamb wave propagates through the eddy. So care should be taken when attributing the fluxes just to the terms in the integrand since the bounds of the integral are changing, which affects the value of the integral. Furthermore the decomposition of Eq. 3 would appear to be quite misleading for this scenario. For instance, the eddy terms could be non-zero since the weak meridional velocity associate with the Lamb wave when added to the velocities in the eddies could give net meridional fluxes, but is this saying anymore than that there are eddies in the same region that an energy flux is occurring?

Many climate models use the hydrostatic balance condition and the governing equations include Lamb waves as solutions. Therefore they may be capable of simulating some of the compressibility effects that have been investigated in fully compressible nonhydrostatic models including propagation of total energy at the speed of sound. However, the use of semi-implicit time differencing schemes, which has become common, brings this into question. Such schemes are considered of great practical utility since they slow down fast moving modes thought to have little meteorological significance, thereby allowing the use of a much larger model time step (e.g. Robert 1969; Tapp and White 1976; Tanguay et al. 1990; Robert 1992; Simmons and Temperton 1996). Typically Lamb waves and gravity waves will be slowed down and distorted by these methods. However, thermally generated Lamb waves and gravity waves do have physically significant effects, so the use of semi-implicit schemes may be causing more inaccuracy than has been recognized in large- scale global models. For instance, if the thermally generated compression wave shown in Fig. 2 were slowed down while maintaining total energy and mass conservation then it would need to be larger in magnitude. Moreover geostrophic adjustment due to the Coriolis force acting on the horizontal velocities associated with Lamb waves would presumably occur over a smaller spatial scale.



Semi-implicit methods may also have an impact on thermally generated gravity waves. A study of orographic gravity waves was carried out by Shutts and Vosper (2011) using the Met Office Unified Model. They found that the semi-implicit off-centring parameter approach used by the model actually damped orographic waves in a global forecast if a large time-step of 15 minutes was used. However, reducing the time-step to 2 minutes allowed the orographic gravity waves to be

adequately simulated. The results of this study raise the possibility that the large time-steps typically used in global models may not only be significantly impacting thermally generated compression waves, but also thermally generated gravity waves. The physical importance of either of these thermally generated fast propagating wave-like disturbances was not recognized at the time semi-implicit schemes were developed. They were designed for the purpose of eliminating the time step constraint imposed by fast moving mechanically produced sound waves, orographic gravity waves, and vertically

propagating gravity waves above cloud tops, which were thought to be relatively unimportant for large scale modelling.

Omission of the term responsible for generating thermal compression waves on the right side of Eq. 15 does not always lead to small mass changes in the model domain as Klemp and Wilhelmson (1978) suggested would be the case for a cloud simulation. For instance, hurricane simulations that include radiation and initially have no clouds will show an increasing mean surface pressure due to net radiative cooling, particularly at night when there is no shortwave warming counteracting

longwave cooling, until enough latent heating occurs in the domain to counter this effect. Moreover inclusion of this term doesn't necessarily mean there will be no mass change in the domain. Fully compressible simulations with diabatic heating in clouds for instance will generate Lamb waves that propagate to the boundaries of the model domain where if there are radiative boundary conditions they may end up being partially reflected. This is because that while radiative boundary conditions are typically set up for the passage of slower moving gravity waves they may not be perfectly reflective to faster

moving waves. So some mass may start to exit the domain. Such mass changes may not be a problem as far as the accurate simulation of typical meteorological processes are concerned, but is something to be aware of.

The effects of thermally generated compression waves have yet to be fully assessed. It is likely that they are ubiquitous in the atmosphere, generated by diabatic heating variations caused by latent heat release, radiative forcing, and surface heating. Compression waves can also be caused by mass inputs and outputs due to water substance conversions such as evaporation

from the surface. Since the amplitudes of these propagating disturbances are very small it might be assumed their effects are negligible, but in this study the case has been made that this is probably incorrect. Potentially the slowing down of thermally generated compression waves and gravity waves due to the use of semi-implicit schemes could be a source of error in climate models, but just how big these errors might be has yet to be evaluated.

Another issue that is brought up by the apparent role of Lamb waves in the transfer of total energy in the atmosphere is the

meaning of the term "heat transfer". It is common to think of heat as being carried from one place to another by the movement of relatively warm air, sometimes referred to as convective heat transfer. As was discussed in section 2.4 this usage is different from how heat is typically defined in thermodynamics as energy in transit from one system to another by thermal interaction. Since the Earth's TOA radiative fluxes have a net input at low latitudes and output at high latitudes, then for an energy balance to occur in the mean then there must be a poleward energy flux, given by the flux terms in equations

(1) and (2), and this has been termed a heat flux. This flux is complicated since it is comprised of fluxes of gravitational potential energy and latent heat, but it does at least have a term called the enthalpy flux, which is considered to be a convective heat flux that results in a transfer of internal energy. Now if a considerable amount of the internal energy is actually being transferred at the speed of sound and doesn't result in significant temperature perturbations as occurs in the numerical experiments in this study, then this further distances total energy transfer from what has been traditionally thought

of to a large degree as a convective heat transfer. Certainly the transfer of total energy that occurs in these experiments with a fully compressible model would not correspond to what would normally be considered a transfer of heat.

Latent heat transport (water vapour) is a quantity that is advected with the wind so that the transfers associated with expansion and compression only applies to fluxes of dry enthalpy and gravitational potential energy. However, the current



view is that if energy is used to evaporate water at some location A and then the vapour is transported by air currents to another location B where it condenses, then energy has been transported from A to B. This is not such a simple thing to conclude, as it may seem, if the condensation produces a Lamb wave that quickly transports the energy hundreds of kilometres away at the speed of sound.

## 6 Conclusions

The first experimental set up in this study examined the response to a convective scale heat source. The fully compressible version of the model showed in the early stages expansion of the heated air and compression of the adjacent environment. A few minutes after the heat source had been turned on a laterally propagating Lamb wave emerged with highest pressure and density anomalies at the surface with a wave front that propagated away at the speed of sound. The reduction of density in the heated region produced buoyancy and a strong updraft developed with compensating subsidence in the nearby environment. This subsidence also propagated in a wave-like manner, but with a slower speed of ~45 m s$^{-1}$. Comparison with the standard version of RAMS that omits the term that involves the material rate of change of virtual potential temperature in Eq. 15, did not show expansion of air in the heated region or compression of the adjacent air. Nevertheless the diagnosed density still decreased in the heated region. A comparison at 15 min showed the thermally generated buoyancy circulation was virtually identical to that of the fully compressible simulation and so for most cloud modelling purposes the terms on the right side of Eq. 15 can be safely neglected as found by Klemp and Wilhelmson (1978).

So far there has been no study that can be said to have definitely observed the low frequency Lamb waves generated by heat release in convective storms. Detection of thermally generated Lamb waves would bolster the conclusions from numerical modelling studies that they exist and could be playing a role in redistribution of total energy and mass. In this study the value of the positive surface pressure perturbation associated with the Lamb wave at a distance of 300 km from the heat source was approximately 0.05 Pa. The strength of the updraft was quite weak, only ~4 m s$^{-1}$, so this would only correspond to weak convection. For the NP2000 study, which simulated a strong convective storm, and included liquid and ice phase microphysics, the magnitude of the pressure perturbations were considerably larger having peak-to-peak values at 400 km from the storm of ~0.6 Pa. The time interval between these positive and negative peaks was ~20 min. Infrasound with periods of tens of seconds have been detected from severe weather (e.g. Bowman and Bedard 1971), however the only study to date that may possibly have detected low frequency Lamb waves generated by latent heat release in convective storms was by Nishida et al. (2014), although they hypothesized the main excitation mechanism for the background Lamb waves to be atmospheric turbulence. As mentioned in section 2.6 the rms amplitude of the Lamb waves they detected was 0.15 Pa, which falls within the range of these fully compressible simulations of convectively produced Lamb waves.

The second experimental set up examined the response to a continent scale heat source applied near the surface. The surface pressure decreased over land and increased over the adjacent ocean. During the nine-hour simulation there was significant lateral transfer of total energy and mass offshore as far as 3000 km from the shoreline. Approximately a third of the total energy increase occurred over land, and two thirds over the ocean. A similar simulation without compression waves showed a decrease of surface pressure over land, but no increase over the ocean. The net total energy did not increase in the domain even though there was a heat input and the net mass decreased. Comparing the horizontal velocities in the sea-breeze circulation there was approximately a 15% difference in their magnitudes. Running the fully compressible model without the Coriolis force resulted in more total energy and mass transfer off shore showing geostrophic adjustment led to significant confinement of the energy and mass. It seems reasonable to consider a horizontal length scale that is analogous to the Rossby radius of deformation where rotational effects become as important as buoyancy effects. For a horizontal length scale where rotational effects become as important as compressibility effects due to thermal expansion in heated regions, which creates Lamb waves propagating at the speed of sound, this would be given by the speed of sound divided by the Coriolis parameter





at 45 degrees latitude for this experiment, approximately 3000 km. Therefore a significant difference would indeed be expected for this large scale heat source experiment between simulations with and without the Coriolis force.

The equator heat source experiment initially created a symmetrical response, but later on as the wave front propagated to latitudes where the Coriolis force became important an interesting asymmetry developed. There was higher surface pressure to the east than the west and this was also reflected in the total energy and mass fields. The equator heat source produced significant meridional fluxes of enthalpy, potential energy and mass. Turning the heat source off at two hours allowed the thermally generated lamb wave to start to separate from the slower moving buoyancy circulation. The Lamb wave had a leading positive lobe of high pressure and a trailing negative lobe, which is a feature of a two-dimensional wave. Examining the vertically summed total energy perturbation field it was clear that the main region of increased energy in the domain in response to the heating was concentrated in the leading lobe of the Lamb wave several thousands of kilometres from the centre of the heat source.

These results reiterate previous results obtained many years ago that Lamb waves are able to transfer significant amounts of total energy at the speed of sound. It would be fair to say these earlier studies have had little impact on interpretations of the large scale total energy transport. This current study, which shows the vertically summed energy fields, should reinforce the hypothesis that Lamb waves could indeed play a role in the large scale transfer of total energy. There have been numerous observational and numerical modelling studies that calculate the meridional atmospheric fluxes of total energy and find a good agreement with the net global energy inputs and outputs (e.g. Masuda 1988; Magnusdottir and Saravanan 1999; Trenberth and Stepaniak 2003; Yang et al. 2015). Therefore on the face of it there doesn't appear to be any problem with the current interpretation of atmospheric energy fluxes, or the ability of state of the art climate models to accurately simulate them. On the hand we would contend that the results of this current study and previous works are unlikely to be fundamentally flawed, and this raises several questions: Are the studies showing considerable transfer of energy by Lamb waves too idealized and this transfer would be limited in a more realistic modelling framework? Do Lamb waves contribute significantly to the energy transport and global models are able to accurately simulate this transport, while perhaps there is a misinterpretation of the physical attribution to some degree? Is the slowing down of Lamb waves due to semi-implicit time differencing not really a substantial problem since it might be causing for the most part just a delayed response? On the other hand is the slowing down of Lamb waves not causing just a delayed response, but also significantly changing the spatial scale of the response as suggested in section 5? Is the slowing down of Lamb waves due to semi-implicit time differencing causing them to have larger amplitude?

Another issue brought up by this study is that it has repercussions for the concept of heat transfer in a fluid. Total energy transfer is typically referred to as heat transfer, yet the mechanism of total energy transfer examined in this work doesn't have much connection to the more common notion of convective heat transfer. Finally, significant transfers of mass occurred at the speed of sound in these numerical experiments. If expansion and compression of air in response to heat sources and sinks leads to large lateral redistributions of mass at the speed of sound this may have consequences for understanding the mass balance that exists in the long term mean in the atmosphere and perhaps seasonal variations.

*Acknowledgements*. This research was primarily supported by Joan M. Nicholls. Support in part was provided by the National Science Foundation, under Grant NSF AGS 1445875. We are extremely grateful to Saurabh Barve for providing computational assistance and appreciate the helpful suggestions from Dr Robert Walko, Stephen Saleeby and Stephen Herbener for modifying the RAMS code to enable double precision.

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



**Table 1**. List of Experiments

| Experiment designation | Description |
|---|---|
| 1A | Convective scale heat source |
| 1B | Convective scale heat source, without compression waves |
| 2A | Continent scale heat source |
| 2B | Continent scale heat source, without compression waves |
| 2C | Continent scale heat source, without the Coriolis force |
| 3A | Equator heat source |
| 3B | Equator heat source, turned off at 2h |




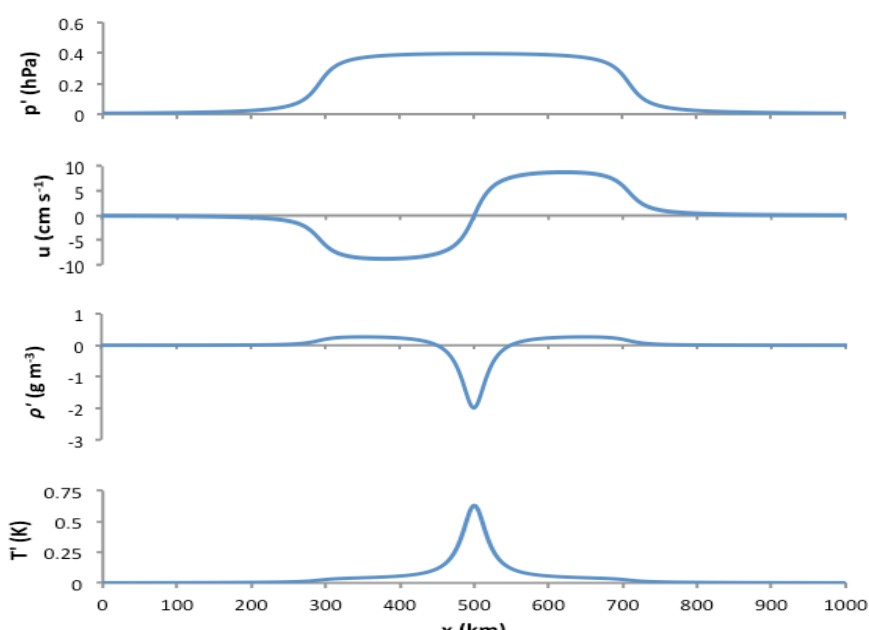

Figure 1. Solution for a 1D thermally generated compression wave at t=600 s.



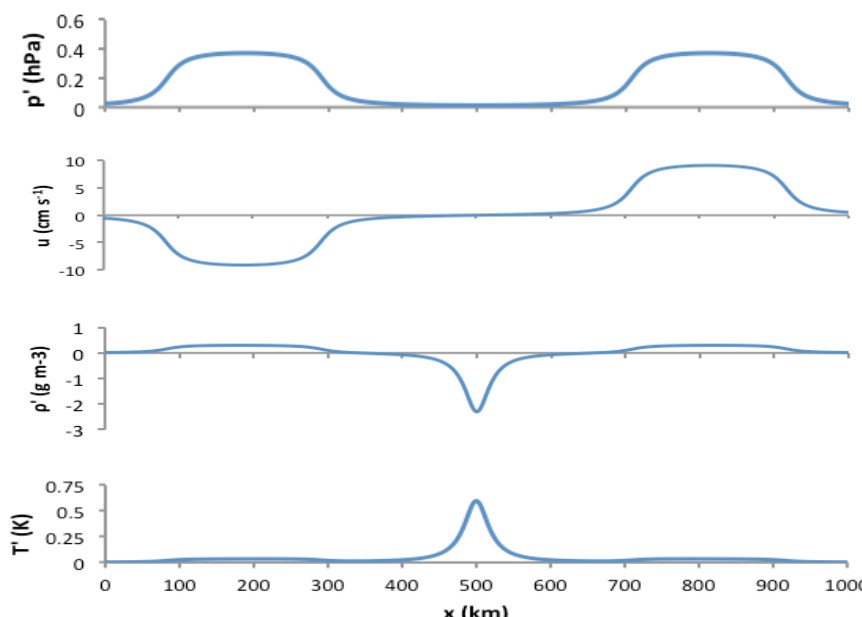

Figure 2. Solution for a 1D thermally generated compression wave at t=1200 s. with the heating turned off at t=600 s.



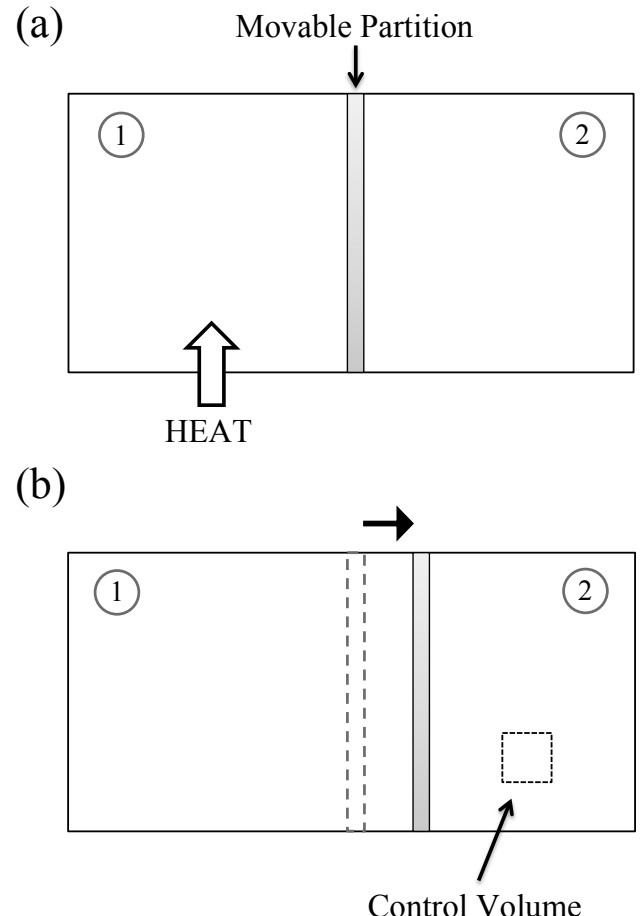

Figure 3. Schematic of a thermally insulated gas divided by a movable partition. (a) Initial state. (b) After heat input in chamber 1.





Figure 4. Vertical sections for the convective scale heat source, Experiment 1A, at t=10 s.
(**a**) Heating rate (J kg$^{-1}$ s$^{-1}$), (**b**) temperature perturbation (K), (**c**) pressure perturbation
(hPa), (**d**) x-component of velocity (m s$^{-1}$). and (**e**) vertical velocity (m s$^{-1}$).





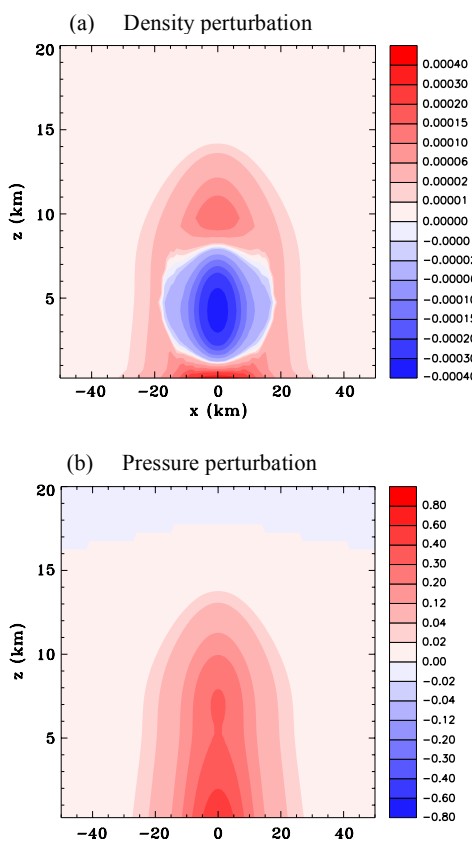

Figure 5. Vertical sections for the convective scale heat source, Experiment 1A, at t=20 s. (**a**) Density perturbation (kg m$^{-3}$), and (**b**) pressure perturbation (hPa).





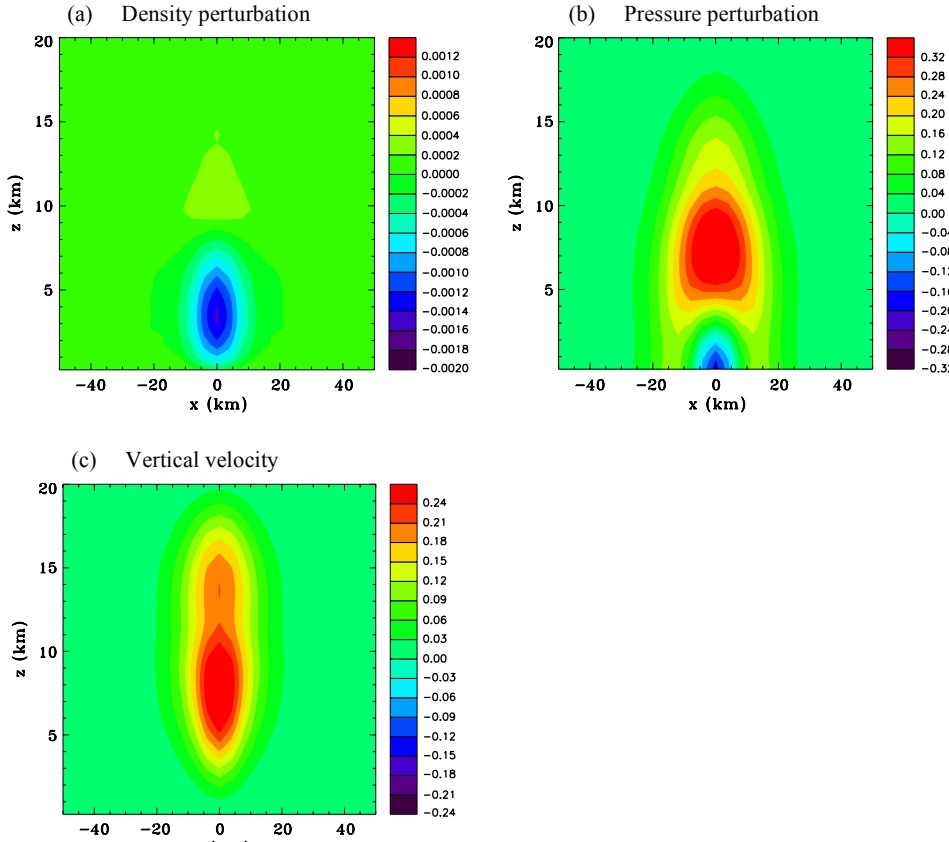

Figure 6. Vertical sections for the convective scale heat source, Experiment 1A, at t=40 s. (**a**) density perturbation (kg m$^{-3}$), (**b**) pressure perturbation (hPa), and (**c**) vertical velocity (m s$^{-1}$).



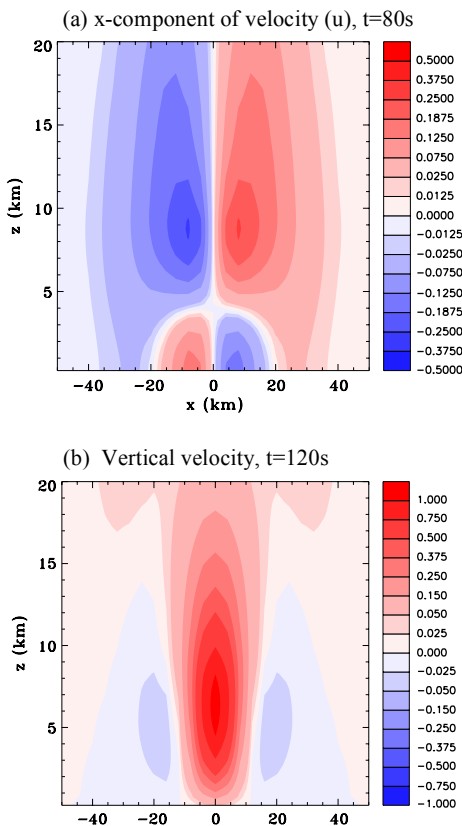

Figure 7. Vertical sections for the convective scale heat source, Experiment 1A. (**a**) x-component of velocity (m s$^{-1}$), at t=80s and (**b**) vertical velocity (m s$^{-1}$), at t=120s.





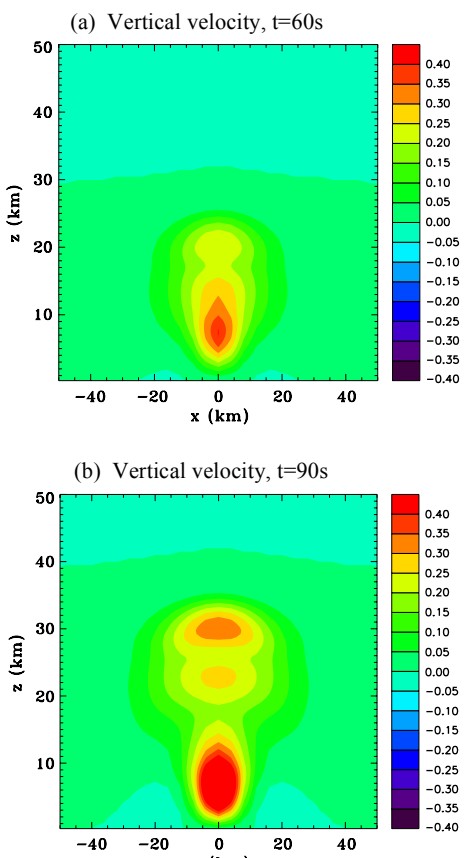

Figure 8. Vertical sections for the convective scale heat source, Experiment 1A. (**a**) Vertical velocity (m s$^{-1}$), at t=60s and (**b**) vertical velocity (m s$^{-1}$), at t=90s.



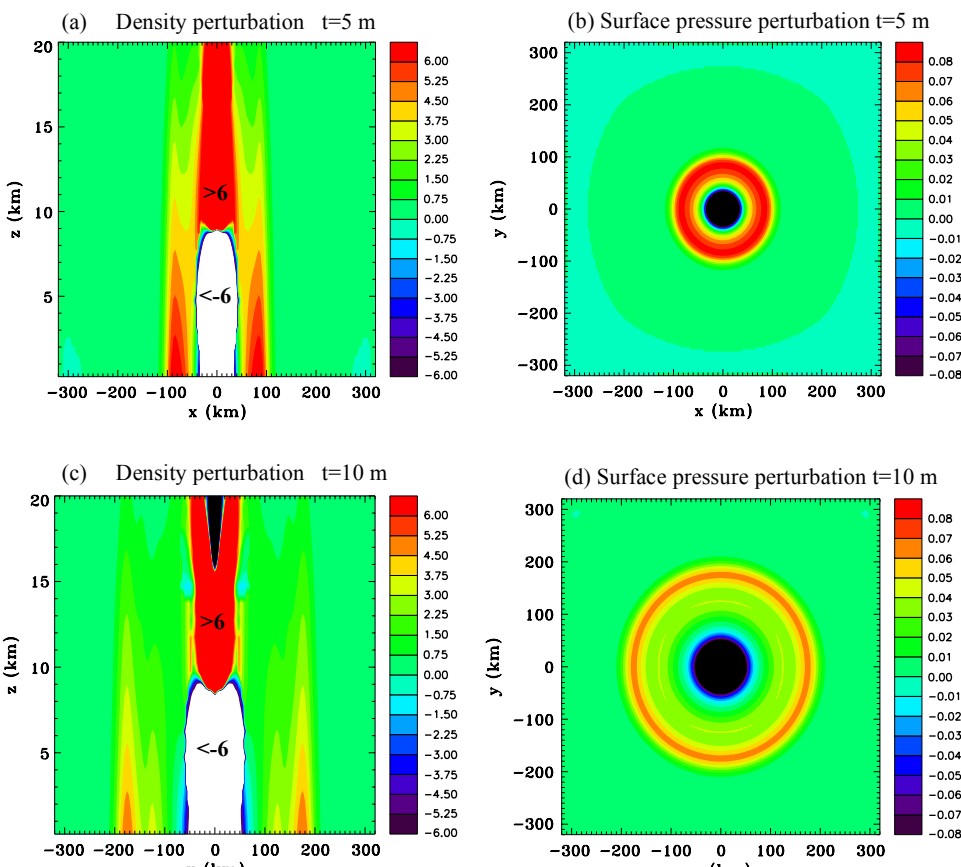

Figure 9. Convective scale heat source, Experiment 1A. Vertical sections of the density perturbation (kg m$^{-3}$ ×10$^{-5}$) and surface pressure perturbation (hPa) at t=5min **(a)** and **(b),** and at t=10 min **(c)** and **(d)**.



Figure 10. Vertical sections for the convective scale heat source, Experiment 1A, at t=15 mins. (**a**) Potential temperature perturbation (K), (**b**) pressure perturbation (hPa), (**c**) density perturbation (kg m$^{-3}$), (**d**) x-component of velocity (m s$^{-1}$), (**e**) vertical velocity (m s$^{-1}$), and (**f**) surface pressure perturbation (hPa).



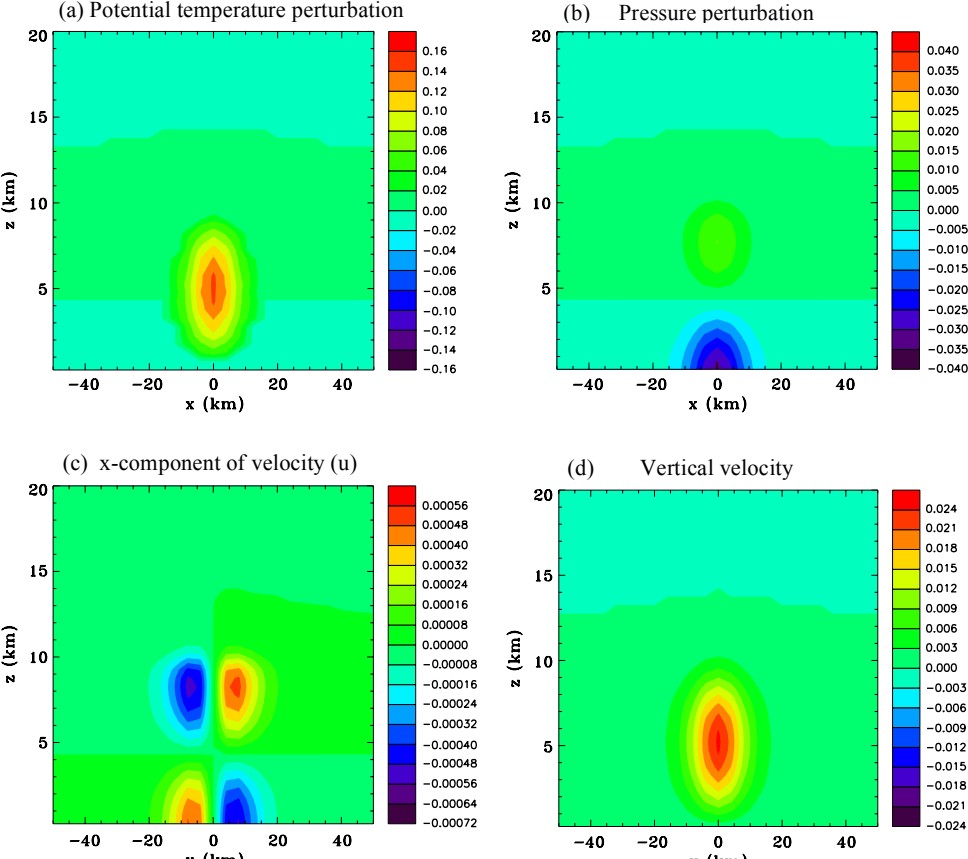

Figure 11. Vertical sections for the convective scale heat source without compression waves, Experiment 1B, at t=10 s. (**a**) Potential temperature perturbation (K), (**b**) pressure perturbation (hPa), (**c**) x-component of velocity (m s$^{-1}$). and (**d**) vertical velocity (m s$^{-1}$).




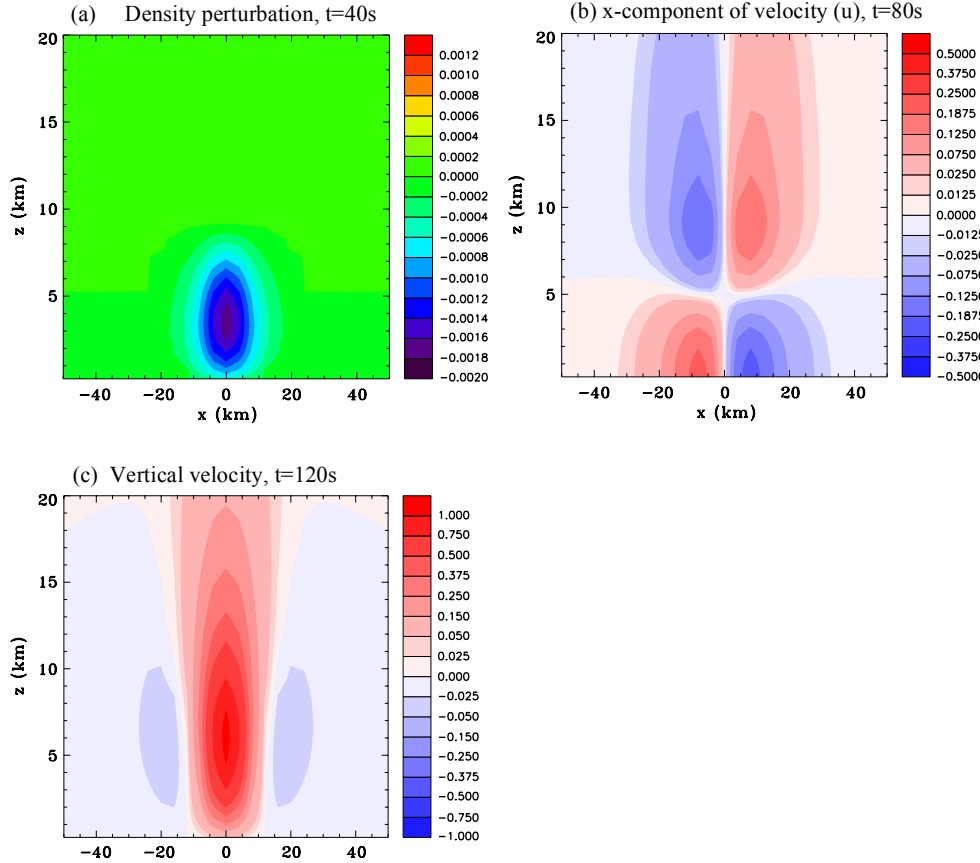

Figure 12. Vertical sections for the convective scale heat source without compression waves, Experiment 1B. (**a**) Density perturbation (kg m$^{-3}$), at t=40s (**b**) x-component of velocity (m s$^{-1}$), at t=80s and (**c**) vertical velocity (m s$^{-1}$), at t=120s.



Figure 13. Vertical sections for the convective scale heat source without compression waves, Experiment 1B, at t=15 minutes. (**a**) Potential temperature perturbation (K), (**b**) pressure perturbation (hPa), (**c**) density perturbation (kg m$^{-3}$), (**d**) x-component of velocity (m s$^{-1}$), and (**e**) vertical velocity (m s$^{-1}$).





Figure 14. Continent-scale heat source, Experiment 2A, at t=9 h. (**a**) Surface potential temperature perturbation (K), (**b**) surface pressure perturbation (hPa), (**c**) vertical section of the density perturbation (kg m$^{-3}$), (**d**) vertical section of the total energy perturbation (J m$^{-3}$), (**e**) vertical summed mass change (kg m$^{-2}$), and (**f**) x-component of velocity (m s$^{-1}$).



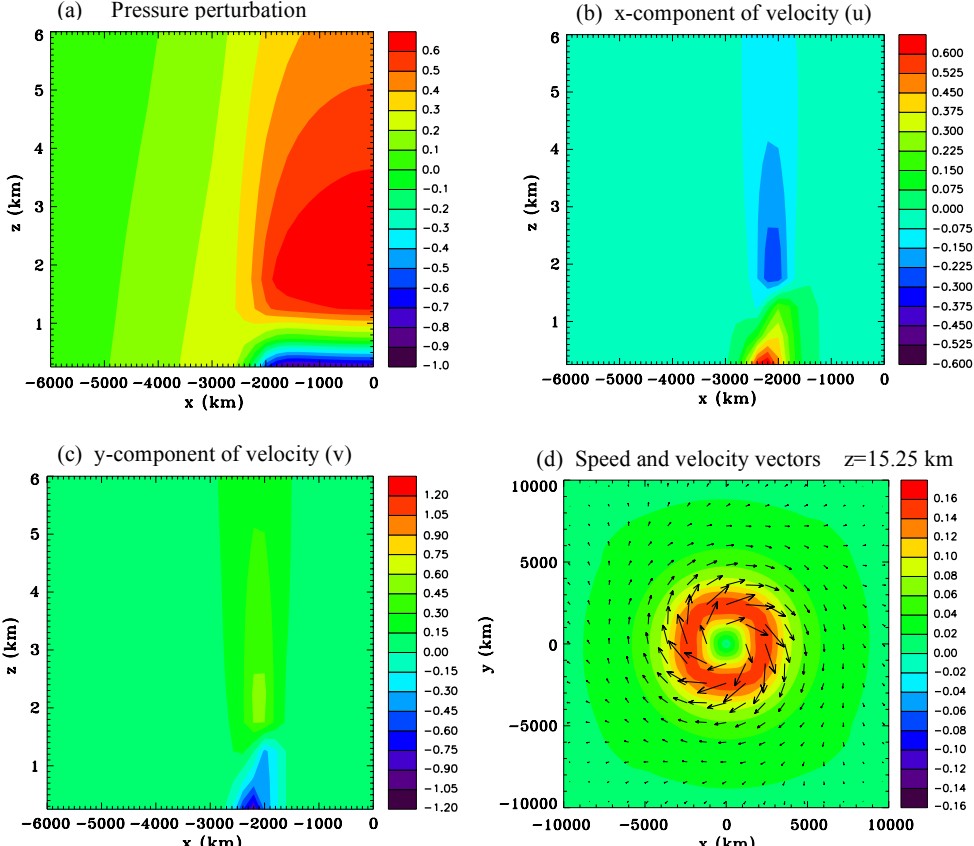

Figure 15. Continent-scale heat source, Experiment 2A, at t=9 h. Zoomed in vertical sections through the centre of the domain of (**a**) pressure perturbation (hPa), (**b**) x-component of velocity (m s$^{-1}$), (**c**) y-component of velocity (m s$^{-1}$), and (**d**) enlarged horizontal section of speed (m s$^{-1}$) and wind vectors at z=15.25 km.





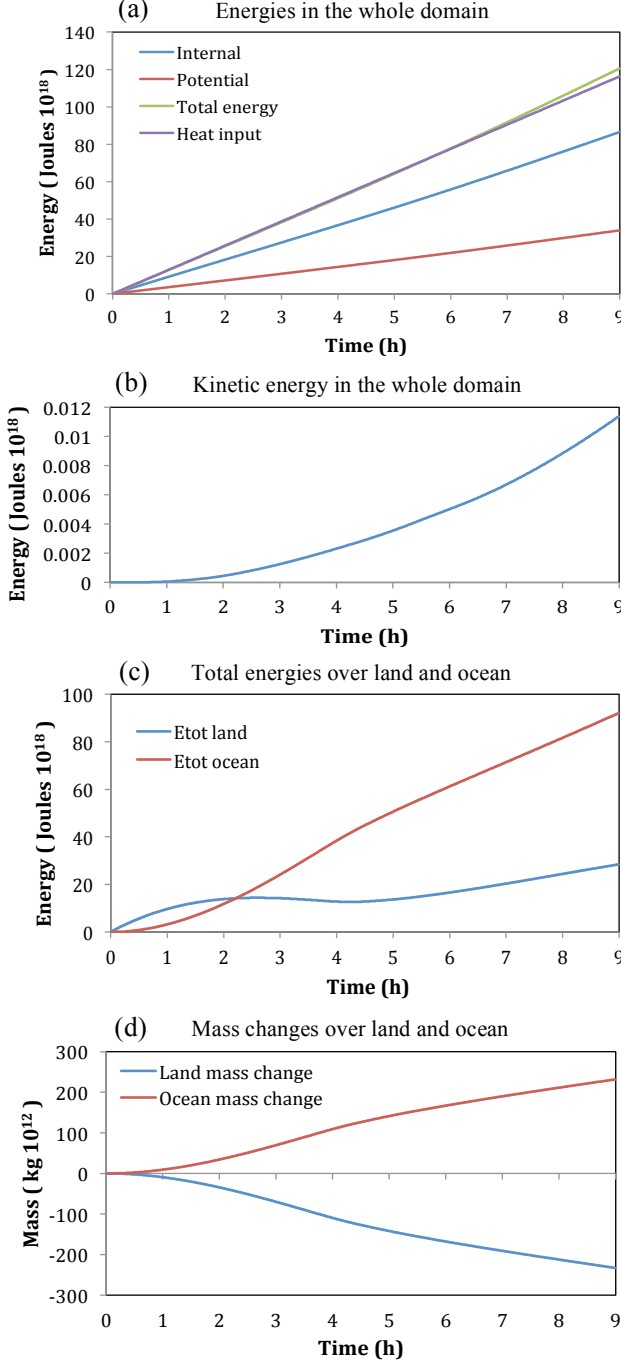

Figure 16. Time series for the continent scale heat source, Experiment 2A. (**a**) Internal energy, potential energy, total energy and heat input, (**b**) kinetic energy, (**c**) energies over land and ocean, (Joules $\times 10^{18}$), and (**d**) mass changes over land and ocean (kg $\times 10^{12}$).





Figure 17. Continent scale heat source without compression waves, Experiment 2B, at t=9 h.
(**a**) Surface potential temperature perturbation (K), (**b**) surface pressure perturbation (hPa), (**c**) vertical section of the density perturbation (kg m$^{-3}$), (**d**) vertical section of the total energy perturbation (J m$^{-3}$), (**e**) vertically summed mass change (kg m$^{-2}$). ), and (**f**) x-component of velocity (m s$^{-1}$).



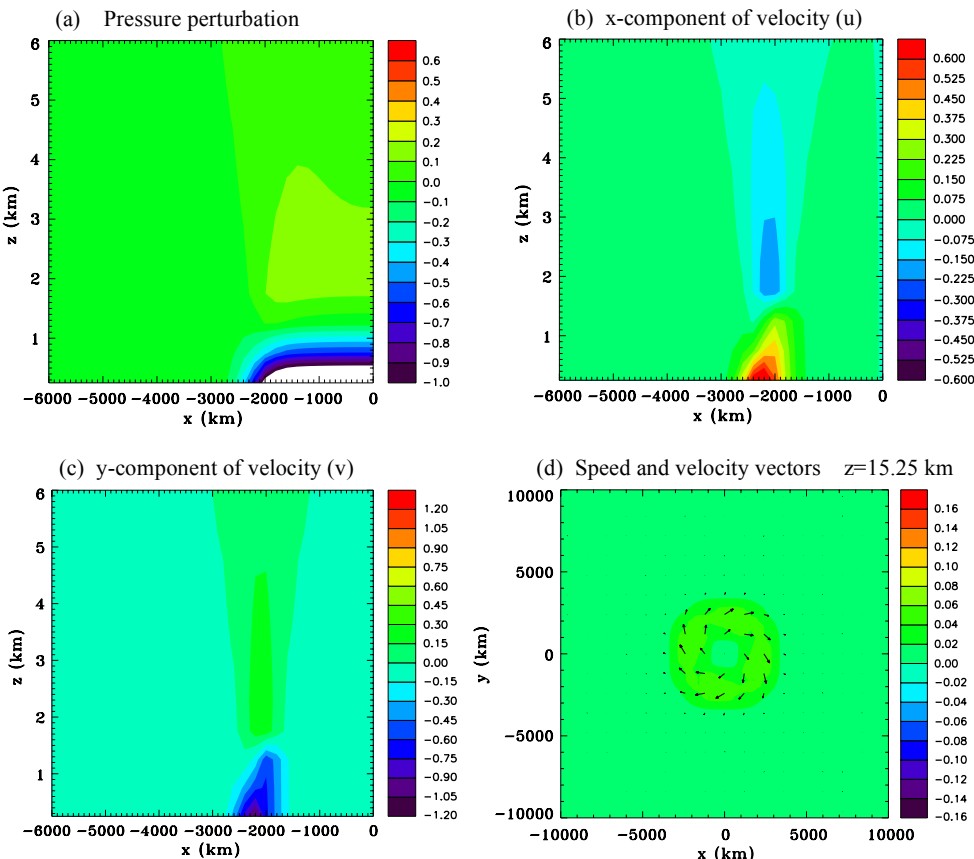

Figure 18. Continent-scale heat source without compression waves, Experiment 2B, at t=9 h. Zoomed in vertical sections through the centre of the domain of (**a**) pressure perturbation (hPa), (**b**) x-component of velocity (m s$^{-1}$), (**c**) y-component of velocity (m s$^{-1}$), and (**d**) enlarged horizontal section of speed (m s$^{-1}$) and wind vectors at z=15.25 km.



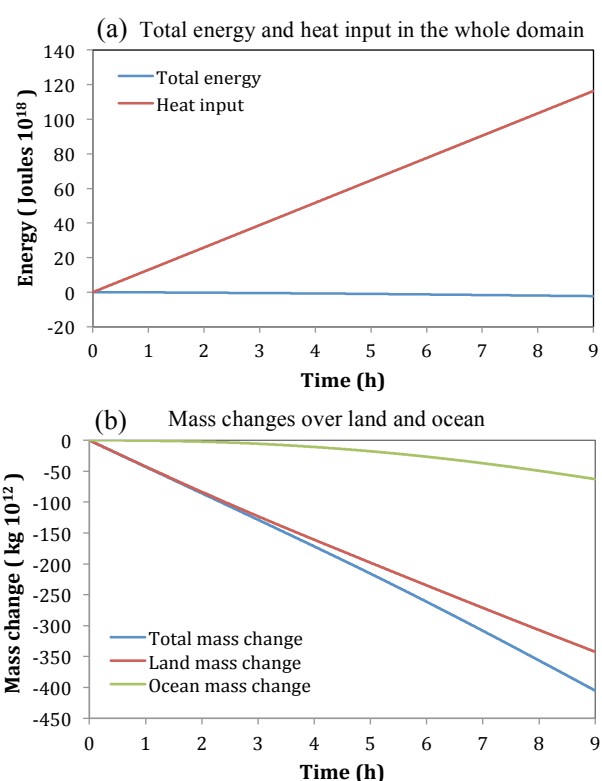

Figure 19. Time series for the continent scale heat source without compression waves, Experiment 2B. (**a**) Total energy and heat input (Joules $\times 10^{18}$), and (**b**) mass changes over land and ocean (kg $\times 10^{12}$).



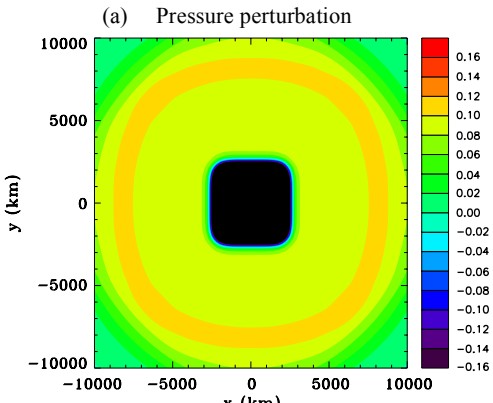

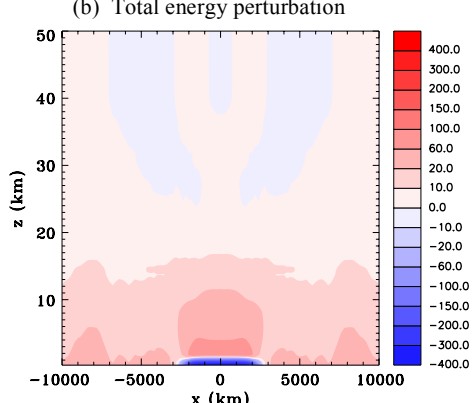

Figure 20. Continent-scale heat source without the Coriolis force, Experiment 2C, at t=9 h. (**a**) Surface pressure perturbation (hPa), and (**b**) vertical section of the total energy perturbation (J m$^{-3}$).



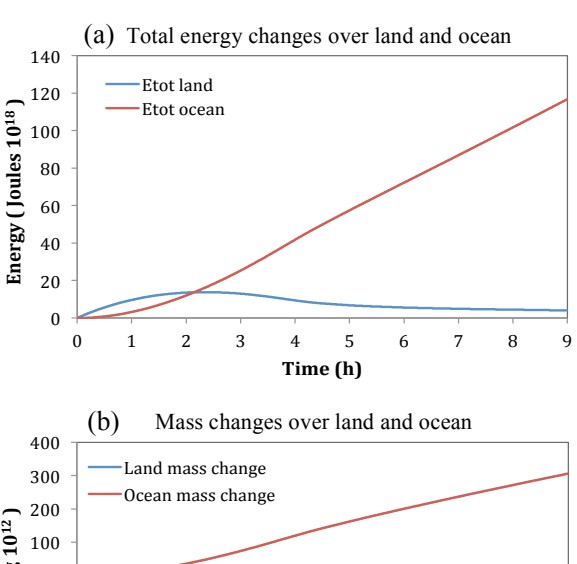

Figure 21. Time series for the square continent scale heating without Coriolis force, Experiment 2C. (**a**) Total energy changes over the land and ocean (Joules $\times 10^{18}$), and (**b**) mass changes over the land and ocean (kg $\times 10^{12}$).





Figure 22. Equator heat source, Experiment 3A, at t=1 h. (**a**) Heating rate (J kg$^{-1}$ s$^{-1}$), (**b**) potential temperature perturbation (K), (**c**) pressure perturbation (hPa), (**d**) x-component of velocity (m s$^{-1}$), (**e**) vertical velocity (m s$^{-1}$), and (f) surface pressure perturbation (hPa).



Figure 23. Equator heat source, Experiment 3A, at t=5 h. (**a**) Potential temperature perturbation (K), (**b**) surface pressure perturbation (hPa), (**c**) density perturbation (kg m$^{-3}$×10$^{-5}$), (**d**) y-component of velocity at the surface (m s$^{-1}$), (**e**) y-component of velocity (m s$^{-1}$), and (**f**) vertical velocity (m s$^{-1}$).





Figure 24. Equator heat source, Experiment 3A, at t=5 h. (**a**) Internal energy perturbation (J m$^{-3}$), (**b**) potential energy perturbation (J m$^{-3}$), (**c**) vertically summed internal energy perturbation (J m$^{-2}$ ×10$^4$), (**d**) vertically summed potential energy perturbation (J m$^{-2}$ ×10$^4$), (**e**) total energy perturbation (J m$^{-3}$), and (**f**) vertically summed mass change (kg m$^{-2}$).





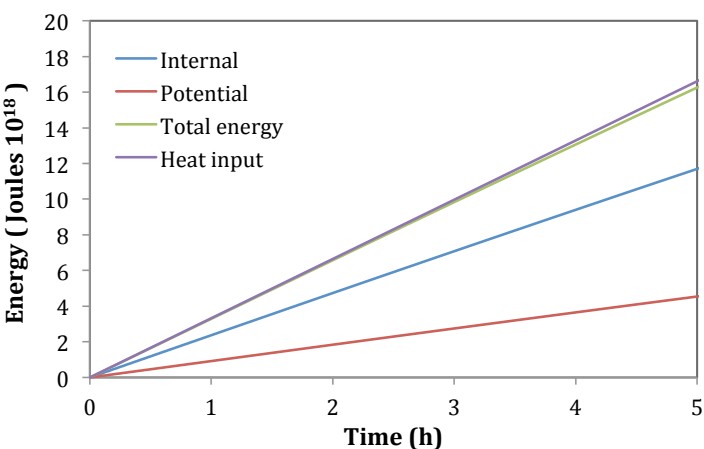

Figure 25. Time series for the equator heat source, Experiment 3A. Internal energy, potential energy, total energy and heat input, (Joules $\times 10^{18}$).



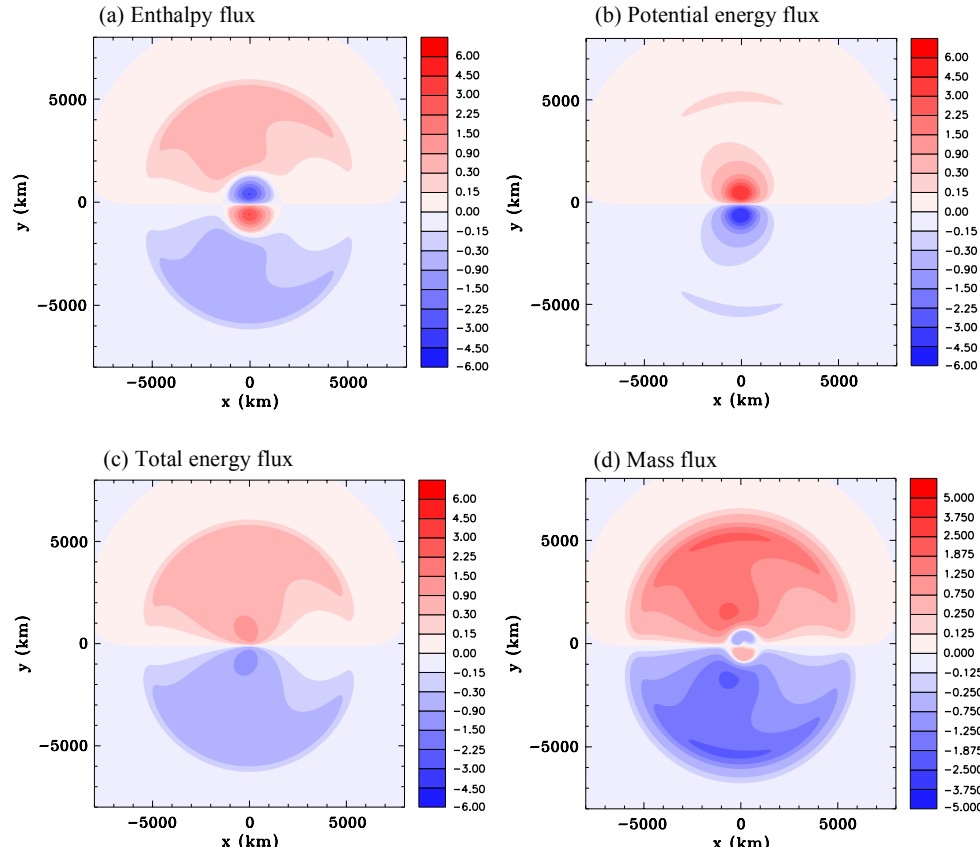

Figure 26. Vertically summed meridional fluxes for the equator heat source, Experiment 3A, at t=5 h. (**a**) Enthalpy flux (W m$^{-1}$ ×10$^8$), (**b**) potential energy flux (W m$^{-1}$ ×10$^8$), (**c**) Total energy flux (W m$^{-1}$ ×10$^8$), and (**d**) mass flux (kg s$^{-1}$ m$^{-1}$ ×10$^2$).



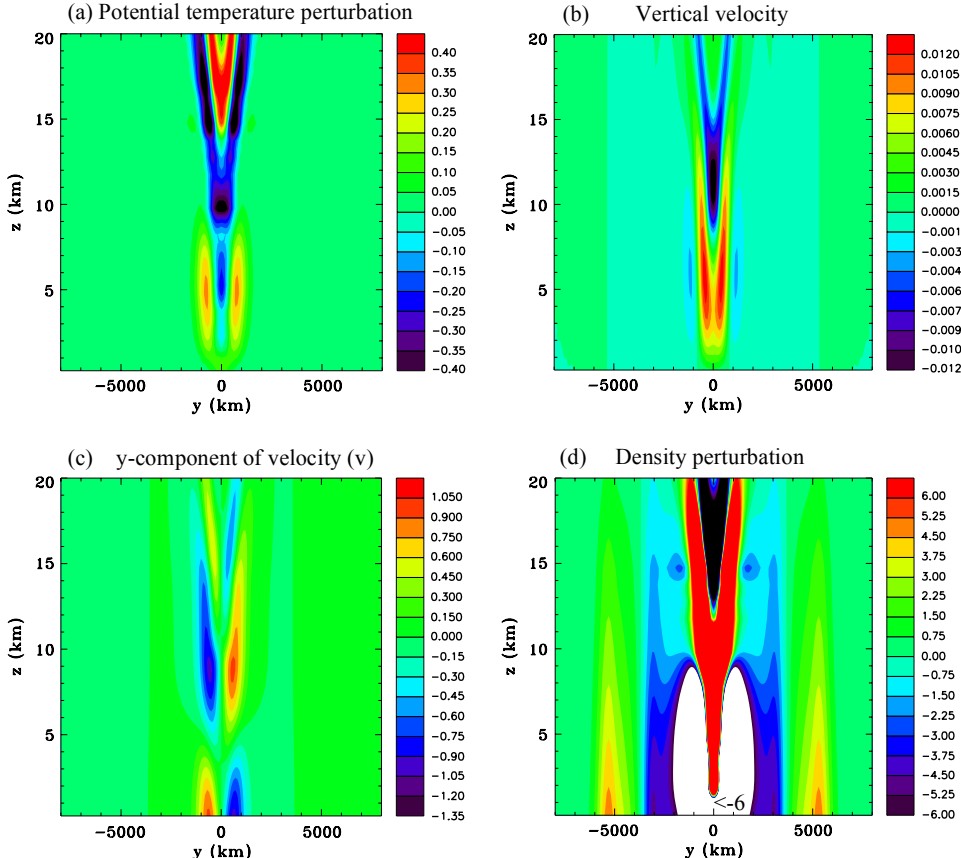

Figure 27. Equator heat source with a duration of 2h, Experiment 3B, at t=5 h. (**a**) Potential temperature perturbation (K), (**b**) vertical velocity (m s$^{-1}$), (**c**) y-component of velocity (m s$^{-1}$), and (**d**) density perturbation (kg m$^{-3}$ ×10$^{-5}$).







Figure 28. Equator heat source with a duration of 2h, Experiment 3B, at t=5 h. (**a**) Internal energy perturbation (J m$^{-3}$), (**b**) potential energy perturbation (J m$^{-3}$), (**c**) total energy perturbation (J m$^{-3}$), (**d**) vertically summed total energy perturbation (J m$^{-2}$ ×10$^{4}$), and (**e**) vertically summed mass perturbation (kg m$^{-2}$).