# Peer review of "On the role of thermal expansion and compression in large scale atmospheric energy and mass transports"

_Atmospheric Chemistry and Physics, 2018_

## Referee Comment (RC1) · Anonymous Referee #1 · 17 May 2018

The authors used idealized simulations to illustrate the importance of thermal compression waves in the total energy transport at the speed of sound (in those simulations). This part is convincing.

The authors also used these results to question the traditional view of energy transport by transient eddy, stationary eddy, and mean meridional circulations. However, their argument is not convincing.

If their claim can be quantified (see the suggestion below), it would represent an important contribution.

Major comments:

[Figure]

Conceptually, eddies in traditional view include waves (e.g., gravity and Rossby waves). Therefore the authors' claim (that the traditional view considers the total energy transfer solely in an advective-like manner by the winds) seems incorrect.

Quantitatively, the authors should show the corresponding results based on the traditional view (i.e., using equations 1-3) in the figures and actually demonstrate the differences of these results versus their results. This would help address the relative importance of thermal compression waves.

The presentation is clear. However, for the key points in the manuscript, the use of 28 figures is not justified. The authors could make good use of supplementary material by moving most of the figures there. This would also increase the readability.

Minor comments:

Equation 2: there should be square brackets on the right hand side, to be consistent with equation 3.

Figures 1 and 2: when u' is as large as 5-10 m/s, should a nonlinear model (rather than a linearized model) be used? How does it affect the results?

Figure 9: is the exponent (-5) in "kg mˆ-3 10ˆ-5" correct in the figure 9 caption?

---

## Referee Comment (RC2) · Anonymous Referee #2 · 19 Jun 2018

Review of

On the role of thermal expansion and compression in large scale atmospheric energy and mass transports

By M. E. Nicholls and R. A. Pielke

The manuscript discusses the role of gravity waves and especially acoustic waves in transporting mass and energy in response to atmospheric disturbances caused by diabatic heating. It makes use of physical arguments, didactic examples, and numerical experiments. These are important ideas, and, given the increasing use of fully compressible equation sets in operational and research models, it is timely to re-emphasize

them. That said, the key ideas involve basic, well-understood fluid dynamics, though they may be obscured by the acoustically filtered equation sets often used in meteorology. Moreover, much of the present manuscript seems to be re-iterating points made in previous work (Nicholls and Pielke 1994a,b, 2000). One of my main points is that the manuscript should be revised to minimize this repetition and to emphasize the new ideas and results.

Main points

1. As mentioned above, repetition of previous work should be minimized and the new results emphasized.

2. It would be useful to relate the ideas and experiments discussed here to the ideas of hydrostatic and geostrophic adjustment, e.g., Gill A. E. (1982), pp 191-, Bannon, P. R., (1995) J. Atmos. Sci., 1743-1752. In particular, Gill (p194) discusses the energy carried away by Poincare waves in the (shallow water) geostrophic adjustment process.

3. The full version of equation (15) involves the initial profiles $\bar{\rho}$ and $\bar{\theta}_v$, so some approximation to the full fluid dynamical equations seems to be involved. This then raises the question of whether the full version of (15) has exact, or only approximate, mass and energy conservation properties. These properties seem to be crucial to the whole exercise, but I wasn't able to quickly track this information down by looking at the references given.

4. The full version of (15) is compared with one that omits the first three terms on the RHS. This omission eliminates the thermal compression waves under study, but at the cost of sacrificing mass and energy conservation. Strong arguments have been made that numerical models, should solve 'dynamically consistent' equation sets, retaining appropriate conservation laws, including mass and energy, even when those equation sets are approximate. For example, the hydrostatic equations and at least some versions of anelastic and pseudo-incompressible equations are dynamically consistent in this sense, and they are widely used. It would therefore be of great interest to understand how such equation sets respond to a local diabatic heating, and how the energy budget is to be interpreted in such models. (It is conceivable that a similar adjustment process occurs to that in the fully compressible case, but instantaneously rather than at the speed of sound.) I believe that a comparison of the full equation set with hydrostatic and/or anelastic equations would be of much wider interest than the artificial system that involves dropping the RHS of (15).

5. P3 lines 16-17: 'This decomposition tacitly makes the assumption...' Actually the decomposition (3) is correct, but the tactit assumption is often made when interpreting (3). I think part of the problem is imprecise use of terminology in the community: the 'transients' in (3) morph into 'eddies' (a bit ambiguous) which morph into 'turbulence' (definitely wrong for the compression waves discussed here). See also section 2.4.

6. Section 2.4. I think part of the problem is that, because of the historical use of acoustically filtered equation sets such as anelastic, or hydrostatic in pressure coordinates, the term 'heat flux' has become identified with the potential temperature flux, which in turn is closely related to the entropy flux. Thus the energy and entropy budgets have become confounded in our thinking. But, while entropy is carried along with the fluid, to a good approximation, energy is not. See also the discussion in Nicholls and Pielke 1994b. If the paper can help to clarify these issues then that would be a useful service to the community.

7. I found the results section hard work. I think it could be better organized to emphasize the points the authors wish to make, and to help the reader to make comparisons between different experiments.

Minor points

P6 line 14: frequency should be period (or change 5 min to 2 pi / 5 min).

P7 line 16: greater than 0?

Section 2.7: I am perfectly happy with the idea that these disturbances are waves,

and for me this section is unnecessary (though perhaps the authors have met some resistance to the idea and so feel that the section is necessary).

---

## Author Comment (AC1) · 30 Jun 2018

Response to referee #1

"Conceptually, eddies in traditional view include waves (e.g., gravity and Rossby waves). Therefore the authors' claim (that the traditional view considers the total energy transfer solely in an advective-like manner by the winds) seems incorrect."

As far as we know, total energy has not been considered to be transported with the phase speed or group velocity of large-scale waves, such as Rossby waves. Rather the meridional component of velocity associated with these waves have been consid-

ered to cause a meridional transfer of total energy. Large wavelength Rossby waves have westward group velocity and so will transfer energy westward, but this is energy associated with the kinetic energy and potential energy of the wave, rather than the quantity total energy. There are however, many quotes that can be found in the literature alluding to total energy transfer in an advective-like manner by the winds. For example:

1) When discussing heat in sensible form and in latent form exchanged between one latitude and another Priestley (1949) states: "The main agents of meridional transport have been assumed to be the broad and deep northward and southward flowing currents of air whose dimensions are comparable in size to the cyclonic and anti-cyclonic pressure systems." This suggests transport at the speed of the air currents.

2) Shaw and Pauluis (2012) state: "Advection by the Eulerian mean meridional circulation dominates the total energy transport in the tropics" and "In the region of the Ferrel cell, a poleward energy transport is achieved by atmospheric eddies, which transport both sensible and latent heat poleward".

3) Liang et al. (2018) state: "In the time mean, the atmosphere carries heat meridionally by moving poleward air parcels with high moist static energy and moving equatorward air parcels with low moist static energy". Carrying heat meridionally by moving air parcels equatorwards and polewards suggests an advective-like process.

The decomposition of Eq. 3 will include the contribution of expansion and compression to the dry static energy flux. It is the physical interpretation of what causes the flux that we think is brought into question by our results.

"Quantitatively, the authors should show the corresponding results based on the traditional view (i.e., using equations 1-3) in the figures and actually demonstrate the differences of these results versus their results. This would help address the relative importance of thermal compression waves."

Our simulations are quite simple in that we prescribe a local heat source and examine a short time later the perturbed energy and mass fields. To examine the traditional view we would have to simulate the Hadley cell, Rossby waves and baroclinic eddies etc. That is a good idea, but goes beyond the scope of this work. We agree with the referee that in questioning the traditional view we are hypothesizing and perhaps we should be more cautious in our statements. However, we do need to say something about how these results may be relevant to the traditional view.

The situation is complicated since observational studies and many modeling studies enforce mass balance when computing poleward total energy transport (Trenberth 1991; Masuda 1988; Keith 1995; Magnusdottir and Saravanan 1999; Graversen et al. 2007; Yang et al. 2015; Liang et al. 2018). This typically involves adding a correction to the meridional velocity (e.g. Yang et al. 2015; see appendix). Our line of thinking is that the larger latent heat release at low latitudes than at high latitudes is causing expansion at low latitudes and compression at higher latitudes, which leads to a poleward mass transfer, and that the resultant buoyancy driven circulations and horizontal eddies are bringing mass back from the high latitudes to the low latitudes to give a mass balance. For instance, an eddy that has warm air moving poleward and cold air equatorward may result in an equatorward mass transfer since the warm air tends to be less dense and the cold air more dense. Both thermal expansion/compression and buoyancy driven circulations are all taking place at the same time, which makes the situation complicated. While there must be a mass balance in the long term mean, transfer of total energy by expansion and compression fundamentally involves mass transfer, so this process might not be accounted for properly if a mass balance condition is imposed. However, this is a hypothesis and at this stage we cannot say much about this issue.

As we stated above the decomposition of Eq. 3 will still include the contribution of expansion and compression to the dry static energy flux. Whether the traditional methodology accurately assesses the contribution due to eddies is unclear to us, particularly

since studies typically employ a mass balance condition. To assess the relative contribution of eddies versus the expansion/compression mechanism would require carefully designed experiments preferably with a fully compressible global model. The referee is correct that we are not able to say how important thermal compression waves are to the total energy transport based on our results. However, given that we find considerable transport of total energy at the speed of sound in our idealized simulations we think the current traditional view is incomplete.

Interestingly a recent paper by Liang et al. (2018) seems to be trying to remove a component involving poleward mass transfer in the total energy transport to come up with a new definition of heat transport. Recognition that considerable mass and total energy transport might be occurring at the speed of sound could provide a different perspective on this problem.

"The presentation is clear. However, for the key points in the manuscript, the use of 28 figures is not justified. The authors could make good use of supplementary material by moving most of the figures there. This would also increase the readability."

We agree that it is a good idea to make use of supplementary material to reduce the number of figures.

Minor comments: "Equation 2: there should be square brackets on the right hand side, to be consistent with equation 3."

Eq. 2 has the factor $2\pi R e \cos(\text{phi})$ which is the distance around the earth at the latitude phi. So that expression on the right hand side is the flux across a circle of latitude. This equation is however usually written as a time average (Starr and White 1954; Oort and Piexóto 1983), although the derivation doesn't require an average in time to be taken. When an average in time is taken there is an approximate mass balance, so the mechanism we are seeing for meridional transfer of total energy that occurs on very short time scales (hours), will not be able to be resolved given the time averaging in observational studies is quite large (days or months). It is possible that by considering

time-averaged fluxes this is causing a misinterpretation of how a considerable portion of the energy transport is physically accomplished. However, this is a hypothesis and would have to be subject to a rigorous study.

"Figures 1 and 2: when u' is as large as 5-10 m/s, should a nonlinear model (rather than a linearized model) be used? How does it affect the results?"

The units are cm/s so the velocity perturbation is quite small.

"Figure 9: is the exponent (-5) in "kg mËȨ-3 10ËȨ-5" correct in the figure 9 caption?"

Yes the exponent is correct.

REFERENCES

Graversen, R. G., E. Kallen, M. Tjernstrom and H. Kornich, 2007: Atmospheric mass-transport inconsistencies in the ERA-40 reanalysis. Q. J. R. Meteorol. Soc., 133, 673-680.

Keith, D. W., 1995: Meridional energy transport: uncertainty in zonal means. Tellus, 47 A, 30-44.

Liang, M., A. Czaja, R. Graversen, and R. Tailleux, 2018: Poleward energy transport: is the standard definition relevant at all time scales? Clim. Dyn., 50, 1785-1797.

Magnusdottir, G., and R, Saravanan, 1999: The response of atmospheric heat transport to zonally averaged SST trends. Tellus, 51, 815-832.

Masuda, K., 1988: Meridional heat transport by the atmosphere and the ocean: analysis of FGGE Data. Tellus, 40A, 285-302.

Oort, A. H., and Peixóto, J. P.: Global angular momentum and energy balance requirements from observations, Advances in Geophysics, 25, 355-490, 1983.

Priestley, C. H. B., 1949: Heat transport and zonal stress between latitudes. Q. J. R. Meteorol. Soc., 75, 28-40.

Shaw, T. A., and A. Pauluis, 2012: Tropical and subtropical meridional latent heat transports by disturbances to the zonal mean and their role in the general circulation. J. Atmos. Sci., 69, 1872–1889.

Starr, V. P., and R. M. White, 1954: Balance requirements of the general circulation. Air Force Cambridge Research Directorate. Geophys. Res. Pap., 35, 1-57.

Trenberth, K. E., 1991: Climate diagnostics from global analysis: conservation of mass in ECMWF analyses. J. Climate, 4, 707-722.

Yang, H., Q. Li, K. Wang, Y. Sun, and D. Sun, 2015: Decomposing the meridional heat transport in the climate system. Clim. Dyn., 44, 2751-2768.

---

## Author Comment (AC2) · 5 Jul 2018

Main points

"1. As mentioned above, repetition of previous work should be minimized and the new results emphasized."

We do think that it is good idea to summarize some of our previous results although perhaps we can shorten this to some degree. Reactions to our previous published work on this topic have on the whole been quite negative. It is clear that the idea that compression waves could be transporting considerable amounts of total energy at the

speed of sound is regarded with a high degree of skepticism. Moreover most climate researchers who study energy transport seem to be unaware of this possibility. We set out in this paper to make a very clear case that there could be a significant large-scale transfer of dry static energy at the speed of sound. We think that presenting the basic ideas more succinctly helps readers to follow the paper without having to read in detail the previous articles. The point is well taken though and we should emphasize what is new about these results. In particular, while it was difficult to modify the model for double precision this has enabled larger scale simulations to be conducted and a good match between energy input and the net change in the total energy field to be achieved. By showing vertically summed energy fields it is apparent that there is transfer occurring at the speed of sound. This is important to establish since large-scale thermal compression waves have not been observed in the atmosphere yet and their existence and effect on energy transport is at this stage purely a prediction based on solutions of the fully compressible Navier-Stokes equations. We have also shown that for a very large-scale heat source there is a fairly small but nevertheless significant difference between a simulation that includes thermal compression waves and one that doesn't. The effects of Coriolis force have been shown to be significant and we have introduced the idea of a Rossby radius of deformation for these fast moving waves. We have shown in detail what happens when there is a convective-scale heat source in a fully compressible model and compared to results for a model that behaves more like an anelastic system. The case has been made that using semi-implicit time differencing methods that slow down thermally generated compression waves and gravity waves might be leading to larger inaccuracies than expected. These new results present a more significant challenge to the traditional view of total energy transport than our previous studies suggesting the current theory is incomplete.

"2. It would be useful to relate the ideas and experiments discussed here to the ideas of hydrostatic and geostrophic adjustment, e.g., Gill A. E. (1982), pp 191-, Bannon, P. R., (1995) J. Atmos. Sci., 1743-1752. In particular, Gill (p194) discusses the energy carried away by Poincare waves in the (shallow water) geostrophic adjustment

process."

This is an interesting suggestion. There are similarities between the analytical thermal compression wave solution and shallow water solutions. The energy discussion by Gill (p194) is based on wave energetics. For the linearized compressible equations a wave energy equation can be derived (e.g. Nicholls and Pielke 1994a, Eq. 36). In that manuscript it was shown in equations (37) and (38) that the wave potential energy is far smaller than the perturbation internal energy in a thermal compression wave. So two energy conservation equations can be derived, one that is wave energy (wave potential energy plus kinetic) and one that is total energy (internal plus kinetic), and the wave energy is far smaller in magnitude. It would be interesting to consider the geostrophic adjustment process for thermally generated compression waves in a more simplified theoretical framework.

"3. The full version of equation (15) involves the initial profiles \bar{\rho} and \bar{\theta}_v, so some approximation to the full fluid dynamical equations seems to be involved. This then raises the question of whether the full version of (15) has exact, or only approximate, mass and energy conservation properties. These properties seem to be crucial to the whole exercise, but I wasn't able to quickly track this information down by looking at the references given."

Equation (15) does indeed use approximations. These are appropriate for regional atmospheric modeling simulations as long as the departure from the basic state is small. For the simulations in this paper these approximations are reasonable and energy conservation is not compromised too much. However if this model was used to simulate large-scale baroclinic eddies then modifications to the model equations would probably need to be made to obtain accurate mass and energy conservation. For instance, there are large meridional potential temperature differences from equator to pole and for a horizontally homogeneous basic state the potential temperature perturbations would be large and the departure from the basic state would not be small.

"4. The full version of (15) is compared with one that omits the first three terms on the RHS. This omission eliminates the thermal compression waves under study, but at the cost of sacrificing mass and energy conservation. Strong arguments have been made that numerical models, should solve 'dynamically consistent' equation sets, retaining appropriate conservation laws, including mass and energy, even when those equation sets are approximate. For example, the hydrostatic equations and at least some versions of anelastic and pseudo-incompressible equations are dynamically consistent in this sense, and they are widely used. It would therefore be of great interest to understand how such equation sets respond to a local diabatic heating, and how the energy budget is to be interpreted in such models. (It is conceivable that a similar adjustment process occurs to that in the fully compressible case, but instantaneously rather than at the speed of sound.) I believe that a comparison of the full equation set with hydrostatic and/or anelastic equations would be of much wider interest than the artificial system that involves dropping the RHS of (15)."

We agree it would be of great interest to compare the full equation set with other approximate equation sets and understand how the energy budget is to be interpreted. As Klemp and Wilhelmson (1978) point out there are similarities of the system Eq. (15) without the terms on the RHS with the anelastic system. There has been considerable interest in the applicability of soundproof models to large-scale motions (e.g. Benacchio et al. 2015). It is not clear to us how other models that are not fully compressible and do not simulate Lamb waves that propagate at the speed of sound would respond to localized heat sources and whether the redistribution of total energy and mass would occur in a realistic manner. We hypothesize that a hydrostatic large-scale model that includes Lamb waves as solutions that are not slowed down would probably be conserving total energy and mass and in a similar way to a fully compressible large-scale model. This is certainly an area that should be investigated further.

"5. P3 lines 16-17: 'This decomposition tacitly makes the assumption...' Actually the decomposition (3) is correct, but the tacit assumption is often made when interpreting

(3). I think part of the problem is imprecise use of terminology in the community: the 'transients' in (3) morph into 'eddies' (a bit ambiguous) which morph into 'turbulence' (definitely wrong for the compression waves discussed here). See also section 2.4."

Yes this is correct and we should reword this sentence.

"6. Section 2.4. I think part of the problem is that, because of the historical use of acoustically filtered equation sets such as anelastic, or hydrostatic in pressure coordinates, the term 'heat flux' has become identified with the potential temperature flux, which in turn is closely related to the entropy flux. Thus the energy and entropy budgets have become confounded in our thinking. But, while entropy is carried along with the fluid, to a good approximation, energy is not. See also the discussion in Nicholls and Pielke 1994b. If the paper can help to clarify these issues then that would be a useful service to the community."

We agree that this is part of problem. It would be interesting to examine the entropy transports. We are not sure that there wouldn't be some entropy transport with the propagation of thermal compression waves. However it appears that a quantity more related to the usual view of "heat flux" can be derived from the equation for entropy, which is basically a potential temperature flux. We took this approach in deriving an approximate conservation equation in Nicholls and Pielke (1994b, Eq. 21). Maybe there is some wider applicability of this approach to large-scale circulations but we have not examined this issue.

"7. I found the results section hard work. I think it could be better organized to emphasize the points the authors wish to make, and to help the reader to make comparisons between different experiments."

We will work on improving the organization of the results.

Minor points

"P6 line 14: frequency should be period (or change 5 min to 2 pi / 5 min)."

We will correct this.

"P7 line 16: greater than 0?"

Thank you for noticing this error.

"Section 2.7: I am perfectly happy with the idea that these disturbances are waves, and for me this section is unnecessary (though perhaps the authors have met some resistance to the idea and so feel that the section is necessary)."

We will think about this section further. We do think that while these disturbances have wave-like characteristics, there are aspects that are not normally associated with waves.

REFERENCES

Bennachio, T., W. P. O'Neill, and R. Klein 2015: A blended soundproof-to-compressible numerical model for small- to mesoscale atmospheric dynamics. Mon. Wea. Rev., 142, 4416-4438.
* * *

---

## Author Response (AR1)

Response to referee #1

**The authors used idealized simulations to illustrate the importance of thermal compression waves in the total energy transport at the speed of sound (in those simulations). This part is convincing.**

**The authors also used these results to question the traditional view of energy transport by transient eddy, stationary eddy, and mean meridional circulations. However, their argument is not convincing.**

**If their claim can be quantified (see the suggestion below), it would represent an important contribution.**

**Major comments:**

**"Conceptually, eddies in traditional view include waves (e.g., gravity and Rossby waves). Therefore the authors' claim (that the traditional view considers the total energy transfer solely in an advective-like manner by the winds) seems incorrect."**

As far as we know, total energy has not been considered to be transported with the phase speed or group velocity of large-scale waves, such as Rossby waves. Rather the meridional component of velocity associated with these waves have been considered to cause a meridional transfer of total energy. Large wavelength Rossby waves have westward group velocity and so will transfer energy westward, but this is energy associated with the kinetic energy and potential energy of the wave, rather than the quantity total energy.

There are however, many quotes that can be found in the literature alluding to total energy transfer in an advective-like manner by the winds. For example:

1) When discussing heat in sensible form and in latent form exchanged between one latitude and another Priestley (1949) states: "The main agents of meridional transport have been assumed to be the broad and deep northward and southward flowing currents of air whose dimensions are comparable in size to the cyclonic and anti-cyclonic pressure systems." This suggests transport at the speed of the air currents.

2) Shaw and Pauluis (2012) state: "Advection by the Eulerian mean meridional circulation dominates the total energy transport in the tropics" and "In the region of the Ferrel cell, a poleward energy transport is achieved by atmospheric eddies, which transport both sensible and latent heat poleward".

3) Liang et al. (2018) state: "In the time mean, the atmosphere carries heat meridionally by moving poleward air parcels with high moist static energy and moving equatorward air parcels with low moist static energy". Carrying heat meridionally by moving air parcels equatorwards and polewards suggests an advective-like process.

We think "soley in an advective-like manner' isn't unreasonable, but we have changed the text in most places so as not to use soley.

**"Quantitatively, the authors should show the corresponding results based on the traditional view (i.e., using equations 1-3) in the figures and actually demonstrate the differences of these results versus their results. This would help address the relative**

**importance of thermal compression waves.”**

Our simulations are quite simple in that we prescribe a local heat source and examine a short time later the perturbed energy and mass fields. To examine the traditional view we would have to simulate the Hadley cell, Rossby waves and baroclinic eddies etc. That is a good idea, but goes beyond the scope of this work. We agree with the referee that in questioning the traditional view we are hypothesizing and in this revised manuscript we have been more cautious in our statements. However, we do need to say something about how these results may be relevant to the traditional view.

 The situation is complicated since observational studies and many modeling studies enforce mass balance when computing poleward total energy transport (Trenberth 1991; Masuda 1988; Keith 1995; Magnusdottir and Saravanan 1999; Graversen et al. 2007; Yang et al. 2015; Liang et al. 2018). This typically involves adding a correction to the meridional velocity (e.g. Yang et al. 2015; see appendix). Our line of thinking is that the larger latent heat release at low latitudes than at high latitudes is causing expansion at low latitudes and compression at higher latitudes, which leads to a poleward mass transfer, and that the resultant buoyancy driven circulations and horizontal eddies are bringing mass back from the high latitudes to the low latitudes to give a mass balance. For instance, an eddy that has warm air moving poleward and cold air equatorward may result in an equatorward mass transfer since the warm air tends to be less dense and the cold air more dense. Both thermal expansion/compression and buoyancy driven circulations are all taking place at the same time, which makes the situation complicated. While there must be a mass balance in the long term mean, transfer of total energy by expansion and compression fundamentally involves mass transfer, so this process might not be accounted for properly if a mass balance condition is imposed. However, this is a hypothesis and at this stage we cannot say much about this issue.

 As we stated above the decomposition of Eq. 3 will still include the contribution of expansion and compression to the dry static energy flux. Whether the traditional methodology accurately assesses the contribution due to eddies is unclear to us, particularly since studies typically employ a mass balance constraint. To assess the relative contribution of eddies versus the expansion/compression mechanism would require carefully designed experiments preferably with a fully compressible global model. The referee is correct that we are not able to say how important thermal compression waves are to the total energy transport based on our results. However, given that we find considerable transport of total energy at the speed of sound in our idealized simulations we think the current traditional view is incomplete.

 Interestingly a recent paper by Liang et al. (2018) seems to be trying to remove a component involving poleward mass transfer in the total energy transport to come up with a new definition of heat transport. Recognition that considerable mass and total energy transport might be occurring at the speed of sound could provide a different perspective on this problem.

 In order to make the limitations of the idealized experiments clearer in the introduction (page 2, line 23) it is now stated:
*“For simplicity this study does not consider water substance. It utilizes an idealized framework, by looking at the transfers of energy and mass that occur in response to imposed heat sources in an initially quiescent atmosphere. Therefore it is difficult to draw definitive conclusions about how much energy and mass might be transferred by this mechanism in a more realistic large scale atmospheric circulation that includes for instance the Hadley circulation, Rossby waves and baroclinic eddies.”*

Also in section 6 the discussion from page 15, line 28 to page 16, line 20, has been significantly modified to address these issues.

**The presentation is clear. However, for the key points in the manuscript, the use of 28 figures is not justified. The authors could make good use of supplementary material by moving most of the figures there. This would also increase the readability.**

We agree that it is a good idea to make use of supplementary material to reduce the number of figures. We have moved eight figures to the Supplement. These are the figures showing the solution for the 1D linearized thermal compression wave, the thought experiment figure, the figures showing the results of the convective-scale heat source with compression waves, and the results for the continent-scale heat source without the Coriolis term.

**Minor comments:**

**Equation 2: there should be square brackets on the right hand side, to be consistent with equation 3.**

Eq. 2 has the factor $2\pi R_e \cos\phi$ which is the distance around the earth at the latitude $\phi$. So that expression on the right hand side is the flux across a circle of latitude. This equation is however usually written as a time average (Starr and White 1954; Oort and Piexóto 1983), although the derivation doesn't require an average in time to be taken. When an average in time is taken there is an approximate mass balance, so the mechanism we are seeing for meridional transfer of total energy that occurs on very short time scales (hours), will not be able to be resolved given the time averaging in observational studies is quite large (days or months). It is possible that by considering time-averaged fluxes this is causing a misinterpretation of how a considerable portion of the energy transport is physically accomplished. However, this is a hypothesis and would have to be subject to a rigorous study.
   It is now stated *"Note that Eqs. 1 and 2 are often expressed as a time average."*

**Figures 1 and 2: when u' is as large as 5-10 m/s, should a nonlinear model (rather than a linearized model) be used? How does it affect the results?**

The units are cm/s so the velocity perturbation is quite small.

**Figure 9: is the exponent (-5) in "kg mˆ-3 10ˆ-5" correct in the figure 9 caption?**

Yes the exponent is correct.

Response to referee #2

**The manuscript discusses the role of gravity waves and especially acoustic waves in transporting mass and energy in response to atmospheric disturbances caused by diabatic heating. It makes use of physical arguments, didactic examples, and numerical experiments. These are important ideas, and, given the increasing use of fully compressible equation sets in operational and research models, it is timely to reemphasize hem. That said, the key ideas involve basic, well-understood fluid dynamics, though they may be obscured by the acoustically filtered equation sets often used in meteorology. Moreover, much of the present manuscript seems to be reiterating points made in previous work (Nicholls and Pielke 1994a,b, 2000). One of my main points is that the manuscript should be revised to minimize this repetition and to emphasize the new ideas and results.**

**Main points**

**"1. As mentioned above, repetition of previous work should be minimized and the new results emphasized."**

In the introduction it is now stated more explicitly what is new about this study:
*"Previous work is extended to larger scales and the effects of including the Coriolis force are examined. Comparison is made between simulations with and without thermally generated compression waves and it is demonstrated that differences start to become noticeable when the horizontal scale of the diabatic forcing is very large. Fields of vertically summed energies and mass are constructed to give a clear perspective of the lateral transports. Furthermore, the total heat input is calculated and compared to the total energy perturbation in the model domain to check energy conservation."*

   Furthermore, it is now worded more clearly in section 2.2 that the 1D linear solution was derived in NP94a and in section 2.3 that the thought experiment was presented in NP94b. Additionally, sections 2.2 and 2.3 have been shortened and the figures moved to the Supplement as well as some of the discussion. We do think it is important to include this material as background material covering the main ideas, since this does not appear a topic that many are familiar with.

**2. It would be useful to relate the ideas and experiments discussed here to the ideas of hydrostatic and geostrophic adjustment, e.g., Gill A. E. (1982), pp 191-, Bannon, P. R., (1995) J. Atmos. Sci., 1743-1752. In particular, Gill (p194) discusses the energy carried away by Poincare waves in the (shallow water) geostrophic adjustment process.**

This is an interesting suggestion. There are similarities between the analytical thermal compression wave solution and shallow water solutions. The energy discussion by Gill (p194) is based on wave energetics. For the linearized compressible equations a wave energy equation can be derived (e.g. Nicholls and Pielke 1994a, Eq. 36). In that manuscript it was shown in equations (37) and (38) that the wave potential energy is far smaller than the perturbation internal energy in a thermal compression wave. So two energy conservation equations can be derived, one that is wave energy (wave potential energy plus kinetic) and one that is total energy (internal plus kinetic), and the wave energy is far smaller in magnitude. It would be interesting to consider the geostrophic adjustment process for thermally generated compression waves in a more simplified theoretical framework.

In the Supplement we now include a comparison of the total energy and wave energy (Supplement Eq. S6 and S7, and Figure S3), and these are mentioned in the main manuscript

(page 4, lines 8-10, and lines 18-20).

**"3. The full version of equation (15) involves the initial profiles \bar{\rho} and \bar{\theta}_v, so some approximation to the full fluid dynamical equations seems to be involved. This then raises the question of whether the full version of (15) has exact, or only approximate, mass and energy conservation properties. These properties seem to be crucial to the whole exercise, but I wasn't able to quickly track this information down by looking at the references given."**

Equation (15, now Eq. 10) does indeed use approximations. These are appropriate for regional atmospheric modeling simulations as long as the departure from the basic state is small. For the simulations in this paper these approximations are reasonable and energy conservation is not compromised too much. However if this model was used to simulate large-scale baroclinic eddies then modifications to the model equations would probably need to be made to obtain accurate mass and energy conservation. For instance, there are large meridional potential temperature differences from equator to pole and for a horizontally homogeneous basic state the potential temperature perturbations would be large and the departure from the basic state would not be small.

This issue is now discussed on page 16, line 16-20.

**"4. The full version of (15) is compared with one that omits the first three terms on the RHS. This omission eliminates the thermal compression waves under study, but at the cost of sacrificing mass and energy conservation. Strong arguments have been made that numerical models, should solve 'dynamically consistent' equation sets, retaining appropriate conservation laws, including mass and energy, even when those equation sets are approximate. For example, the hydrostatic equations and at least some versions of anelastic and pseudo-incompressible equations are dynamically consistent in this sense, and they are widely used. It would therefore be of great interest to under-stand how such equation sets respond to a local diabatic heating, and how the energy budget is to be interpreted in such models. (It is conceivable that a similar adjustment process occurs to that in the fully compressible case, but instantaneously rather than at the speed of sound.) I believe that a comparison of the full equation set with hydrostatic and/or anelastic equations would be of much wider interest than the artificial system that involves dropping the RHS of (15)."**

We agree it would be of great interest to compare the full equation set with other approximate equation sets and understand how the energy budget is to be interpreted. As Klemp and Wilhelmson (1978) point out there are similarities of the system Eq. (15, now Eq. 10)) without the terms on the RHS with the anelastic system. There has been considerable interest in the applicability of soundproof models to large-scale motions (e.g. Benacchio et al. 2015). It is not clear to us how other models that are not fully compressible and do not simulate Lamb waves that propagate at the speed of sound would respond to localized heat sources and whether the redistribution of total energy and mass would occur in a realistic manner. We hypothesize that a hydrostatic large-scale model that includes Lamb waves as solutions that are not slowed down would probably be conserving total energy and mass and in a similar way to a fully compressible large-scale model. This is certainly an area that should be investigated further.

**5. P3 lines 16-17: 'This decomposition tacitly makes the assumption...' Actually the decomposition (3) is correct, but the tacit assumption is often made when interpreting (3). I think part of the problem is imprecise use of terminology in the community: the 'transients'**

in (3) morph into 'eddies' (a bit ambiguous) which morph into 'turbulence' (definitely wrong for the compression waves discussed here). See also section 2.4.**

Yes this is correct and we have reworded this sentence (page 3, line 25):
*"A tacit assumption that is commonly made in interpreting Eq. 3 is that total energy is a quantity that is advected solely with the wind"*
We have also added on page 3, line 31-34.
*"There is nothing intrinsically wrong with the decomposition in Eq. 3 and the transfer of energy by compression waves would be included. However, the assumption that total energy is transported in an advective manner may have led to practical methodologies for computing the contributions in Eq. 3 that do not account for the possibility of energy transport at the speed of sound. This issue will be discussed further in section 6."*

Also in section 6 the discussion from page 15, line 28 to page 16, line 20, has been significantly modified.

**"6. Section 2.4. I think part of the problem is that, because of the historical use of acoustically filtered equation sets such as anelastic, or hydrostatic in pressure coordinates, the term 'heat flux' has become identified with the potential temperature flux, which in turn is closely related to the entropy flux. Thus the energy and entropy budgets have become confounded in our thinking. But, while entropy is carried along with the fluid, to a good approximation, energy is not. See also the discussion in Nicholls and Pielke 1994b. If the paper can help to clarify these issues then that would be a useful service to the community."**

We agree that this is part of problem. It would be interesting to examine the entropy transports. We are not sure that there wouldn't be some entropy transport with the propagation of thermal compression waves. However it appears that a quantity more related to the usual view of "heat flux" can be derived from the equation for entropy, which is basically a potential temperature flux. We took this approach in deriving an approximate conservation equation in Nicholls and Pielke (1994b, Eq. 21). Maybe there is some wider applicability of this approach to large-scale circulations but we have not examined this issue. We don't think we can address the issue of entropy flux in this paper unfortunately.

**"7. I found the results section hard work. I think it could be better organized to emphasize the points the authors wish to make, and to help the reader to make comparisons between different experiments."**

We have moved eight figures to the Supplement. These are the figures showing the solution for the 1D linearized thermal compression wave, the thought experiment figure, the figures showing the results of the convective-scale heat source with compression waves, and the results for the continent-scale heat source without the Coriolis term. These results have been summarized in the main paper.

**Minor points**

**"P6 line 14: frequency should be period (or change 5 min to 2 pi / 5 min)."**

We have corrected this.

**"P7 line 16: greater than 0?"**

Thank you for noticing this error.

**"Section 2.7: I am perfectly happy with the idea that these disturbances are waves, and for me this section is unnecessary (though perhaps the authors have met some resistance to the idea and so feel that the section is necessary)."**

We think that while these disturbances have wave-like characteristics, there are aspects that are not normally associated with waves so that it is worth keeping this section.

REFERENCES

Bennachio, T., W. P. O'Neill, and R. Klein 2015: A blended soundproof-to-compressible numerical model for small- to mesoscale atmospheric dynamics. *Mon. Wea. Rev*., 142, 4416-4438.

[revised manuscript text omitted]

Melville Nicholls 8/19/18 11:53 AM

Melville Nicholls 8/19/18 11:53 AM
Melville Nicholls 8/19/18 11:53 AM

Melville Nicholls 8/19/18 12:06 PM
Melville Nicholls 8/19/18 12:07 PM

Melville Nicholls 8/19/18 12:07 PM
Melville Nicholls 8/19/18 12:07 PM
Melville Nicholls 8/19/18 12:07 PM
Melville Nicholls 8/19/18 12:08 PM
Melville Nicholls 8/19/18 12:08 PM

Melville Nicholls 8/19/18 12:08 PM
Melville Nicholls 8/19/18 12:09 PM

Melville Nicholls 8/19/18 12:09 PM

Melville Nicholls 8/19/18 2:47 PM
Melville Nicholls 8/19/18 2:47 PM

**5.3 Equator heat source**

In this section the response to a heat source with the scale of a cloud cluster and which is located at the equator is examined (Experiments 3A . Figure 14 shows a vertical section through the centre of the heat source and several panels of simulated fields at 1 h for Experiment 3A. There are quite a few similarities to the convective-scale heat source. A compression wave is generated which at this early stage is quite symmetrical as can be seen by the surface pressure perturbation. Figure 15,
5   with vertical sections that only go to a height of 20 km, shows that by 5 h a significant asymmetry has developed as the compression wave has travelled far enough away from the equator to become influenced by the Coriolis force. The potential temperature has increased to just over one degree in the centre of the heat source. The warm region has widened due to the compensating subsidence (Fig. 15f). The compression wave front has reached about 6000 km from the source as can be seen
10  in Fig. 15b, c and d. It appears that as the wave front reaches higher latitudes the Coriolis force acts to turn the weak outflow to the right in the northern hemisphere and to the left in the southern hemisphere resulting in a convergence of mass to the east of the heat source and a divergence of mass to the west, and this results in an asymmetrical surface pressure field. For instance, Fig. 15d shows that the meridional component of the outflow is stronger to the west of the heat source than to the east. The height-meridional vertical section of the density field shown in Fig. 15c, which has a very small colour bar interval
15  to portray the compression wave, indicates mass redistribution towards the polar-regions. Figs. 15e and f indicate that the leading edge of the thermally generated buoyancy circulation is propagating at a speed of ~45 m s$^{-1}$ in the meridional direction.

[revised manuscript text omitted]

---

## Author Response (AR2)

Response to referee #1

**Comments:**

**The authors have satisfactorily addressed my concerns. My minor concern is: Abstract is too long (with 460 words). It should be shortened by half. This will actually make the Abstract more effective. For instance, the experiment designs could be removed. Furthermore,the impacts of this study could be summarized in a sentence at the end of Abstract, such as "These results raise important questions related to the mass constraints when calculating meridional energy transports, the use of semi-implicit in large scale global models, and the use of heat transfer for total energy transfer."**

The abstract has now been shortened considerably.

\

Response to referee #2

**Comments**

**I feel there is still much to be clarified on this topic, along the lines of points 2 (relation to balance and adjustment), 4 (how would the results look using a fully mass and energy conserving numerical model), and 6 (the relation between energy and entropy budgets), in my original review. Nevertheless, the authors have addressed the most problematic points in the original manuscript, and I think the work is of sufficient interest to justify publication. I point out below a couple of small points that ought to be addressed.**

We agree that there is still much that needs to be clarified on this subject, but these issues are probably best addressed in future studies. We think that the results we have obtained would look virtually identical if some of the approximations in the derivation of the governing equations were removed, since the approximations are reasonable for the magnitude of the perturbations that occur in the simulations. As the reviewer points out the relation between energy and entropy budgets is an important topic that needs to be examined.

**1. Wave energy (p4 line 19, also reply to point 2 of my original review). For a compressible fluid the wave energy should include the potential, internal, and kinetic contributions (e.g., Lighthill 1978, Phillips 1990). For an acoustic wave (i.e., compression wave) the internal contribution is significant and it does not make sense to discuss the `wave energy' without including that contribution.**

This is an interesting point, but having looked at Phillips (1990) we cannot see an equation that contains potential, internal and kinetic contributions. Eq. 2.18 of Phillips (1990) for instance has kinetic energy, wave potential energy and a term arising from the inclusion of gravity (or the thermobaric term). Note in our equations S1-S3 of the Supplement, gravity is not included so this third term does not arise. We are distinguishing between total energy and wave energy. For the simple linearized 1D equations, Eqs. S1-S3, there are two energy conservation equations that can be derived. Eq. S6 is an approximation to the perturbation total energy equation that would be obtained if the full system of equations hadn't been linearized. It is missing a kinetic energy term, but this term is really small compared to the perturbation internal energy term. So if the solution to this linearized system was compared to that of the nonlinear system the solutions would be very similar for small perturbations, and the internal energy anomaly, which is almost identical to the total energy anomaly, would be seen to be propagating away at the speed of sound.
 The other energy conservation equation is called the wave energy equation, which can be derived from Eqs. S1-S3, and is shown in Eq. S7. This is the standard wave energy equation for acoustic waves (e.g. Phillips, 1990, Eq. 2.18), but as mentioned above doesn't have the term arising from including gravity. Fig. S3 compares these two energy expressions showing the wave energy is four orders of magnitude less than the internal energy anomaly.
 These two conservation equations are distinct. The internal energy perturbation equation (Eq. S6) could be added to the wave energy equation (Eq. S7), and possibly this is what the reviewer is suggesting. This would result in a conservation equation for the sum of wave energy and perturbation internal energy, but it is not clear to us how his equation would be interpreted, and moreover the internal energy perturbation would be four orders of magnitude larger than the kinetic and wave potential energy terms. We think it is reasonable to consider these conservation equations independently from one another.
 We now reference Philips (1990) in the supplement, which provides more background material for the reader on wave energy.

**2. P13 line 39, P18 line 30, The idea of a Rossby radius associated with the Lamb wave is well known. For example, it is mentioned in Gill (1982) p207.**

This is correct and we now reference Gill (1982) on pages 13:
"The length scale given by the speed of sound divided by the Coriolis parameter at 45 degrees latitude is approximately 3000 km. For this f-plane simulation this length scale can be considered to be a Rossby radius of deformation for these very fast moving compression waves (e.g. Gill 1982)."

Also on page 18:
"Running the fully compressible model without the Coriolis force resulted in much more total energy and mass transfer offshore, showing that geostrophic adjustment when the Coriolis force is included leads to significant confinement of the energy and mass. For this simulation, which has a constant Coriolis parameter, it is helpful to envisage a Rossby radius of deformation for Lamb waves (e.g. Gill 1982), which is analogous to the Rossby radius of deformation for gravity waves where rotational effects become as important as buoyancy effects. This is given by the speed of sound divided by the Coriolis parameter, which for this experiment is approximately 3000 km. This Rossby radius is too large for the f-plane approximation to be valid (Gill 1982), so while it is useful for understanding the results of this particular experiment it has limited utility when the Coriolis parameter more realistically varies with latitude."

**On the role of thermal expansion and compression in large scale atmospheric energy and mass transports**

Melville E. Nicholls[1] and Roger A. Pielke Sr.[1]

[1]Cooperative Institute for Research in Environmental Sciences, Department of Atmospheric and Oceanic Sciences, University of Colorado, Boulder, CO 80309, USA

*Correspondence to*: Melville E. Nicholls (Melville.Nicholls@colorado.edu)

**Abstract.** There are currently two views of how atmospheric total energy transport is accomplished. One view considers total energy as a quantity that is transported in an advective-like manner by the wind. The other considers that thermal expansion and the resultant compression of the surrounding air causes a transport of total energy in a wave-like manner at the speed of sound. This latter view emerged as the result of detailed analysis of fully compressible mesoscale model simulations that demonstrated considerable transfer of internal and gravitational potential energy at the speed of sound by Lamb waves. In this study, results are presented of idealized experiments with a fully compressible model designed to examine the large-scale transfers of total energy and mass when local heat sources are prescribed. For simplicity a Cartesian grid was used, there was a horizontally homogeneous basic state, and the simulations did not include moisture.

Three main experimental designs were employed: The first has a convective storm scale heat source. The second experiment has a continent scale heat source prescribed near the surface to represent surface heating and includes a constant Coriolis parameter. The third experiment has a cloud cluster scale heat source prescribed at the equator and includes a latitude dependent Coriolis parameter. Results show considerable amounts of meridional total energy and mass transfer at the speed of the sound, which suggests that the current theory of large scale total energy transport is incomplete. Comparison of simulations with and without thermally generated compression waves show that for a very large scale heat source there are fairly small but nevertheless significant differences of the wind field.

These results raise important questions related to the mass constraints when calculating meridional energy transports, the use of semi-implicit time differencing in large scale global models, and the use of the term "heat transfer" for "total energy transfer".

[revised manuscript text omitted]
 much more total energy and mass transfer offshore, showing that geostrophic adjustment when the Coriolis force is included leads to significant confinement of the energy and mass. For this simulation, which has a constant

Coriolis parameter, it is helpful to envisage a Rossby radius of deformation for Lamb waves (e.g. Gill 1982), which is analogous to the Rossby radius of deformation for gravity waves where rotational effects become as important as buoyancy effects. This is given by the speed of sound divided by the Coriolis parameter, which for this experiment is approximately 3000 km. This Rossby radius is too large for the f-plane approximation to be valid (Gill 1982), so while it is useful for understanding the results of this particular experiment it has limited utility when the Coriolis parameter more realistically varies with latitude.

The equator heat source experiment initially created a symmetrical response, but later on as the wave front propagated to latitudes where the Coriolis force became important an interesting asymmetry developed. There was higher surface pressure to the east than the west and this was also reflected in the total energy and mass fields. The equator heat source produced significant meridional fluxes of enthalpy, potential energy and mass. Turning the heat source off at two hours allowed the thermally generated lamb wave to start to separate from the slower moving buoyancy circulation. The Lamb wave had a leading positive lobe of high pressure and a trailing negative lobe, which is a feature of a two-dimensional wave. Examining the vertically summed total energy perturbation field it was clear that the main region of increased energy in the domain in response to the heating was concentrated in the leading lobe of the Lamb wave several thousands of kilometres from the centre of the heat source.

These results reiterate previous results obtained many years ago that Lamb waves are able to transfer significant amounts of total energy at the speed of sound. It would be fair to say these earlier studies have had little impact on interpretations of the large scale total energy transport. This current study, which shows the vertically summed energy fields, should reinforce the hypothesis that Lamb waves could indeed play a role in the large scale transfer of total energy. There have been numerous observational and numerical modelling studies that calculate the meridional atmospheric fluxes of total energy and find a good agreement with the net global energy inputs and outputs (e.g. Masuda 1988; Magnusdottir and Saravanan 1999; Trenberth and Stepaniak 2003; Yang et al. 2015). Therefore on the face of it there doesn't appear to be any problem with the current interpretation of atmospheric energy fluxes, or the ability of state of the art climate models to accurately simulate them. On the hand we would contend that the results of this current study and previous works are unlikely to be fundamentally flawed, and this raises several questions: Are the studies showing considerable transfer of energy by Lamb waves too idealized and this transfer would be limited in a more realistic modelling framework? Do Lamb waves contribute significantly to the energy transport and global models are able to accurately simulate this transport, while perhaps there is a misinterpretation of the physical attribution to some degree? Is the slowing down of Lamb waves due to semi-implicit time differencing not really a substantial problem since it might be causing for the most part just a delayed response? On the other

Melville Nicholls 9/26/18 3:04 PM

Melville Nicholls 9/23/18 4:26 PM

Melville Nicholls 9/23/18 4:00 PM

Melville Nicholls 9/23/18 4:01 PM

Melville Nicholls 9/26/18 3:07 PM

Melville Nicholls 9/26/18 3:09 PM

Melville Nicholls 9/26/18 3:08 PM

[revised manuscript text omitted]